# Constrained Stochastic Nonconvex Optimization with State-dependent Markov Data

Abhishek Roy[*]     Krishnakumar Balasubramanian[†]     Saeed Ghadimi[‡]

## Abstract

We study stochastic optimization algorithms for constrained nonconvex stochastic optimization problems with Markovian data. In particular, we focus on the case when the transition kernel of the Markov chain is state-dependent. Such stochastic optimization problems arise in various machine learning problems including strategic classification and reinforcement learning. For this problem, we study both projection-based and projection-free algorithms. In both cases, we establish that the number of calls to the stochastic first-order oracle to obtain an appropriately defined $\epsilon$-stationary point is of the order $\mathcal{O}(1/\epsilon^{2.5})$. In the projection-free setting we additionally establish that the number of calls to the linear minimization oracle is of order $\mathcal{O}(1/\epsilon^{5.5})$. We also empirically demonstrate the performance of our algorithm on the problem of strategic classification with neural networks.

## 1   Introduction

We consider the following stochastic optimization problem

$$\operatorname*{argmin}_{\theta \in \Theta} f(\theta) = \operatorname*{argmin}_{\theta \in \Theta} \mathbb{E}\left[F(\theta; x)\right], \tag{1}$$

where (i) the expectation is taken over the stationary distribution, $\pi_\theta$, of the random vector $x$, (ii) $F$ (and hence $f$) is a potentially non-convex function in $\theta$, and (iii) $\Theta$ is a compact and convex constraint set. Stochastic approximation algorithms for solving problem (1), given an independent and identically distributed (`iid`) data stream $\{x_k\}_k$ drawn from $\pi$, are well-studied. Such `iid` assumptions are commonly made in various machine learning and statistical problems including empirical risk minimization [SSBD14], sparse recovery [BJMO12] and compressed sensing [FR13, Lan20]. We refer to [MB11, ABRW12, RSS12, GL13, SZ13, LZ16, ACD+19] for a partial list of non-asymptotic upper and lower bounds on the oracle complexity of widely-used stochastic approximation algorithms like the Stochastic Gradient Descent (SGD) and the Stochastic Conditional Gradient Algorithm.

Our focus in this work is on the case when the data sequence $\{x_k\}_k$ is drawn from a Markov chain with a state-dependent transition kernel $P_\theta$. Such a setting arises in several machine learning applications including but not limited to strategic classification [HMPW16, CDP15, MDPZH20, LW22] and reinforcement learning [Bar92, GSK13, ZJM21, KMMW19, QW20]. Despite their prevalence in practice, a deeper understanding of the non-asymptotic oracle complexity of stochastic approximation for Markovian data is only now starting to emerge. We establish non-asymptotic

[*] `abroy@ucdavis.edu`. Halıcıoğlu Data Science Institute, University of California, San Diego. Work done while being affiliated with the Department of Statistics, UC Davis. Research of this author was supported by National Science Foundation (NSF) grant CCF-1934568.

[†] `kbala@ucdavis.edu`. Department of Statistics, University of California, Davis. Research of this author was supported in part by UC Davis CeDAR (Center for Data Science and Artificial Intelligence Research) Innovative Data Science Seed Funding Program and NSF Grant DMS-2053918.

[‡] `sghadimi@uwaterloo.ca`. Department of Management Sciences, University of Waterloo. Research of this author was supported by an NSERC Discovery Grant.

oracle complexity results for the stochastic conditional gradient algorithm for non-convex constrained stochastic optimization with Markovian data. To establish our results, from a methodological point-of-view, we leverage the moving-average stochastic gradient estimation technique recently used in [ZSM$^+$20, GRW20, XBG22] in the context of constrained optimization with iid data. This technique avoids having to a use a mini-batch of samples in each iteration, which turns out to be crucial in the non-iid setup we consider. From a theoretical point-of-view, we assume the so-called drift conditions, a classical assumption in Markov Chain literature [AMP05]. This ensures the existence of a solution to the Poisson equation associated with the underlying Markov chain [DMPS18] which enables one to decompose the noise present in the stochastic gradient into three components: a martingale difference sequence, a time-decaying sequence, and a telescopic sum type sequence. The key idea of our paper is to use this decomposition to construct an auxiliary sequence of iterates with a time-decaying noise-variance and show that these sequence of iterates are *close* to the iterates of the original sequence produced by our algorithm. This novel technique in then used in combination with a merit-function based analysis to establish the oracle complexity results.

## 1.1 Motivating Example

Problems of the form in (1) arise in various important applications, e.g., strategic classification, and reinforcement learning as mentioned above. Below we illustrate the motivation of this work through the example of strategic classification with adapted best response [LW22]. In strategic classification, there is a *learner* whose task is to classify a given dataset which is collected from a set of *agents*. Given the knowledge of the classifier, the agents can distort some of their personal features, in order to get classified in a predetermined target class. This scenario arises in various applications, e.g., spam email filtering, and credit score classification. Optimizing the classifier to classify such strategically modified data where the agents modify the data iteratively can be formulated as problem (1).

Formally, let the classifier be $h(x, \theta)$ where $x \in \mathbb{R}^d$ is the feature and $\theta$ is the parameter to be optimized. $h(x; \cdot) : \Theta \to \mathbb{R}$ is potentially nonconvex. Let the loss function be logistic loss which for a sample $(x, y)$, where $y \in \{-1, 1\}$ denotes the corresponding class, is given by,

$$L(\theta; x, y) = \log\left(1 + \exp\left(-h(x; \theta)\right)\right) + (1 - y)h(x; \theta)/2. \tag{2}$$

We use $x_S$, and $x_{-S}$ to denote the subset of feature $x$ which are respectively strategically modifiable, and non-modifiable by the agents. Then the modified feature (the best response) $x'_S$ reported by the agent is the solution to the following optimization problem:

$$x'_S = \operatorname*{argmax}_{x_S}\left(h(x; \theta) - c(x_S, x'_S)\right), \tag{3}$$

where $c(x, x')$ is the cost of modifying $x_S$ to $x'_S$. Let the agents iteratively learn $x'_S$ similar to [LW22]. Note that unlike [LW22], where the authors deploy a logistic regression classifier and the closed form solution of the best response is readily known to the agents, it may not be the case in general. In that case the agents have to possibly learn the best response $x'_S$ using some iterative optimization algorithm. For example, if the agents use Gradient Ascent then, at every iteration $k$, a set $\mathcal{I}_k$ of $n_1 \leq M$ randomly chosen agents out of $M$ agents modify their features as:

$$x^k_{S,i} = \begin{cases} x^{k-1}_{S,i} + \alpha\left(\nabla h(x^{k-1}_{S,i}; \theta_k) - \nabla c(x^{k-1}_{S,i}, x^0_{S,i})\right) & i \in \mathcal{I}_k \\ x^{k-1}_{S,i} & i \notin \mathcal{I}_k \end{cases} \tag{4}$$

where $\alpha$ is the stepsize. With a little abuse of notation, we use $\nabla h(x^{k-1}_{S,i}; \theta)$ in (4) to denote the fact that the gradient is with respect to $x^{k-1}_{S,i}$ while $x_{-S,i}$ remains unchanged. This introduces the state-dependent Markov chain dynamics in the training data. The objective function, analogous to $f(\theta)$ in (1), is

$$\min_{\theta \in \Theta} \mathbb{E}_{\pi_\theta}\left[L(\theta; x, y)\right],$$

where $\pi_\theta$ is the stationary joint distribution of $(x, y)$, and $\Theta$ is a convex and compact set, e.g., sparsity inducing constraint $\|\theta\|_1 \leq R$ from some $R > 0$. The loss evaluated at a single data point $(x, y)$, $L(\theta; x, y)$, is analogous to $F(\theta; x)$ in (1). [DX20], and [LW22] study this problem theoretically and empirically respectively in an unconstrained strongly convex setting. Our results takes a step towards analyzing this problem in constrained nonconvex setting. We empirically show the performance of the stochastic conditional gradient algorithm on a strategic classification problem in Section 4.1.

## 1.2 Preliminaries and Main Contributions

Before we present our main contributions, we introduce our convergence criterion. In constrained optimization literature, most commonly used convergence criteria are: (i) *Gradient Mapping* (GM), and (ii) *Frank-Wolfe Gap* (FW-gap). The *Gradient Mapping* at a point $\bar{\theta} \in \Theta$ is defined as

$$\mathcal{G}_{\Theta}(\bar{\theta}, \nabla f(\bar{\theta}), \beta) := \beta \left( \bar{\theta} - \Pi_{\Theta} \left( \bar{\theta} - \frac{1}{\beta} \nabla f(\bar{\theta}) \right) \right), \tag{5}$$

where $\Pi_{\Theta}(x)$ denotes the orthogonal projection of the vector $x$ onto the set $\Theta$, i.e.,

$$\Pi_{\Theta} \left( \bar{\theta} - \frac{1}{\beta} \nabla f(\bar{\theta}) \right) = \operatorname*{argmin}_{y \in \Theta} \left\{ \langle \nabla f(\bar{\theta}), y - \bar{\theta} \rangle + \frac{\beta}{2} \|y - \theta\|_2^2 \right\}.$$

We will use $\Pi_{\Theta}(\bar{\theta}, \nabla f(\bar{\theta}), \beta)$ to denote $\Pi_{\Theta} \left( \bar{\theta} - \nabla f(\bar{\theta})/\beta \right)$ when there is no confusion. Note that when $\Theta \equiv \mathbb{R}^d$ we have $\mathcal{G}_{\Theta}(\bar{\theta}, \nabla f(\bar{\theta}), \beta) = \nabla f(\bar{\theta})$. In other words, for constrained optimization gradient mapping plays an analogous role of the gradient for unconstrained optimization. The gradient mapping is a frequently used measure in the literature as a convergence criterion for nonconvex constrained optimization [Nes18]. We should emphasize here that although the gradient mapping cannot be computed in the stochastic setting, one can still use it as a convergence measure.

[BG22] shows that the above notion of convergence criterion is closely related to the so-called *Frank-Wolfe Gap*. The FW-gap is defined as

$$g_{\Theta}(\bar{\theta}, \nabla f(\bar{\theta})) := \max_{y \in \Theta} \langle \nabla f(\bar{\theta}), \bar{\theta} - y \rangle. \tag{6}$$

The following proposition from [BG22] establishes the relation between the gradient mapping criterion and the Frank-Wolfe gap:

**Proposition 1.1** [BG22] *Let $g_{\Theta}(\cdot)$ be the Frank-Wolfe gap defined in (6) and $\mathcal{G}_{\Theta}(\cdot)$ be the gradient mapping defined in (5). Then, we have*

$$\|\mathcal{G}_{\Theta}(\bar{\theta}, \nabla f(\bar{\theta}), \beta)\|^2 \le g_{\Theta}(\bar{\theta}, \nabla f(\bar{\theta})), \qquad \forall \bar{\theta} \in \Theta.$$

*Moreover, under standard regularity assumption in smooth optimization (specifically, Assumption 2.1, and 2.2), we have*

$$g_{\Theta}(\bar{\theta}, \nabla f(\bar{\theta})) \le L \left\| \mathcal{G}_{\Theta}(\bar{\theta}, \nabla f(\bar{\theta}), \beta) \right\|_2 / \beta. \tag{7}$$

In this work we use a suboptimality measure, closely related to both GM and the FW-gap. At point $\bar{\theta} \in \Theta$, we define the suboptimality measure $V(\bar{\theta}, z) : \mathbb{R}^d \times \mathbb{R}^d \to \mathbb{R}$ as [GRW20]

$$V(\bar{\theta}, z) := \left\| \Pi_{\Theta} \left( \bar{\theta} - z/\beta \right) - \bar{\theta} \right\|_2^2 + \left\| z - \nabla f(\bar{\theta}) \right\|_2^2, \tag{8}$$

where $z$, formally defined in Algorithm 1, is the moving-average estimate of $\nabla f(\bar{\theta})$. We show the relation among $V(\theta, z)$, and GM $\mathcal{G}_{\Theta}(\theta, z, \beta)$ in the following proposition.

**Proposition 1.2** *Let $\{z_k\}$ be the sequence generated in Algorithm 1. Then, for $k = 1, 2, \cdots, N$, we have $\|\mathcal{G}_{\Theta}(\theta_k, z_k, \beta)\|_2^2 \le \max(2, 2\beta^2) V(\theta_k, z_k)$.*

The proof is provided in Appendix A. The main objective of this work is to find an $\epsilon$-stationary solution to (1), where an $\epsilon$-stationary solution is defined as follows:

**Definition 1** *A point $\bar{\theta}$ is said to be an $\epsilon$-stationary solution to (1), if $\mathbb{E}\left[ V(\bar{\theta}, z) \right] \le \epsilon$, where the expectation is taken over all the randomness involved in the problem.*

For stochastic Frank-Wolfe-type algorithms, the oracle complexity is measured in terms of number of calls to the Stochastic First-order Oracle (SFO) and the Linear Minimization Oracle (LMO) used to the solve the sub-problems of the algorithm which involves minimizing a linear function over the convex constraint set. Formally, we have the following definition.

**Definition 2** *For a given point $\theta \in \Theta$, SFO returns the stochastic gradient $\nabla F(\theta, x)$. Given a vector $z$, LMO returns a vector $v := \operatorname{argmin}_{y \in \Theta} \langle z, y \rangle$.*

| | | iid | | non-iid | | | |
| | | | | State-independent MC | | State-dependent MC | |
| Algorithm | Criterion | SFO | LMO | SFO | LMO | SFO | LMO |
| --- | --- | --- | --- | --- | --- | --- | --- |
| 1-SFW [ZSM$^+$20] | FW-gap | $\mathcal{O}\left(\epsilon^{-3}\right)$ | $\mathcal{O}\left(\epsilon^{-3}\right)$ | ✗ | ✗ | ✗ | ✗ |
| (ASA+ICG) [XBG22] | GM | $\mathcal{O}\left(\epsilon^{-2}\right)$ | $\mathcal{O}\left(\epsilon^{-3}\right)$ | ✗ | ✗ | ✗ | ✗ |
| (ASA+ICG) [This paper] | GM | | | $\tilde{\mathcal{O}}\left(\epsilon^{-2}\right)$ | $\tilde{\mathcal{O}}\left(\epsilon^{-3}\right)$ | $\mathcal{O}\left(\epsilon^{-2.5}\right)$ | $\mathcal{O}\left(\epsilon^{-5.5}\right)$ |

Table 1: Oracle complexity of projection-free one-sample stochastic conditional gradient algorithms for constrained non-convex optimization, to find an $\epsilon$-stationary point.

Hence, in this work, the oracle complexity is measured in terms of the number of calls to SFO and LMO required by the proposed algorithm to obtain an $\epsilon$-stationary solution as in Definition 1. With the above preliminaries, we now list our **main contributions**:

- In Theorem 3.1, we show that the number of calls to the SFO and LMO required by the stochastic conditional gradient-type method in Algorithm 1, with *state-dependent* Markovian data, is of order $\mathcal{O}(\epsilon^{-2.5})$ and $\mathcal{O}(\epsilon^{-5.5})$ respectively. To the best of our knowledge, these are the first oracle complexity results for projection-free one-sample stochastic optimization algorithm for constrained nonconvex optimization in the Markovian setting.
- In Theorem 3.2, for the sake of completion, we also show that the number of calls to the SFO and LMO required for the case of *state-independent* Markovian data is of the order $\tilde{\mathcal{O}}(\epsilon^{-2})$ and $\tilde{\mathcal{O}}(\epsilon^{-3})$ respectively. In particular, this turns out to be of the same order as that of `iid` data ignoring the logarithmic factors.

A summary of the our contributions is provided in Table 1. We also empirically evaluate our algorithm on a strategic classification problem with 2-layer neural network classifier and show that the proposed method obtains encouraging results. We provide an experiment on single-index model regression with sparsity-inducing nuclear-norm ball constraint in Appendix 4.2.

## 1.3 Related Work

**Stochastic Optimization with Dependent Data.** Understanding stochastic approximation algorithms like SGD with dependent data in the asymptotic setting has been well-explored in the optimization literature. We refer to [KY03, Bor09, BMP12] for a text-book introduction to such classical results. A few recent results include [AMP05, TD17]. In the unconstrained non-asymptotic setting, [DAJJ12] studies convex optimization with ergodic data sequence. [DL22] uses multi-level gradient estimator and analyze AdaGrad for nonconvex optimization with Markovian Data. Block coordinate descent with homogeneous Markov chain has been analyzed in [SSXY20] for nonconvex unconstrained optimization. [DX20] studies stochastic optimization with decision-dependent data distribution for strongly convex functions in the context of strategic classification.

Sample-average approximation algorithms for constrained convex optimization with $\phi$-mixing data was considered in [WPT$^+$21]. [SSY18], and [AL22] analyze projected SGD for constrained non-convex optimization with time-homogeneous Markov chain. None of these works consider state-dependent data distribution except [DX20]. But unlike [DX20], we consider constrained nonconvex optimization. There also exists work in the reinforcement learning literature on understanding stochastic optimization with Markovian data; see, for example [XXLZ20, BRS18, DNPR20]. However, such works are invariably focused on specific objective functions arising in the reinforcement learning setup, while our focus is on obtaining results for a general class of functions.

**Conditional Gradient-Type Method.** There has been significant recent advancements in the conditional gradient algorithm literature although it was developed long back [FW56, LP66]; see [Mig94, Jag13, LJJ15, LJJ15, HJN15, GKS21, BS17], for a non-exhaustive list of recent works. [HK12, HL16] provided expected oracle complexity results for stochastic conditional gradient algorithm in the stochastic convex setup. Better rates were provided by a sliding procedure in [LZ16]. In the non-convex setting, [RSPS16, YSC19, HL16] considered variance reduced stochastic conditional gradient algorithms, and provided expected oracle complexities. [QLX18] analyzed the sliding algorithm in the non-convex setting and provided results for the gradient mapping criterion. All of the above works use increasing orders of mini-batch based gradient-estimate.

To avoid mini-batches, a moving-average gradient estimator based on only one-sample in each iteration for a stochastic conditional gradient-type algorithm was proposed in [MHK20] and [ZSM$^+$20] for the convex and non-convex setting. However, several restrictive assumptions have been made in

[MHK20] and [ZSM$^+$20]. Specifically, [ZSM$^+$20] requires that the stochastic gradient $G_1(x, \xi_1)$ has uniformly bounded function value, gradient-norm, and Hessian spectral-norm, and the distribution of the random vector $\xi_1$ has an absolutely continuous density $p$ such that the norm of the gradient of $\log p$ and spectral norm of the Hessian of $\log p$ has finite fourth and second-moments respectively. In contrasts, we do not require such stringent assumptions.

## 2 Assumptions

We now introduce the precise assumptions we make in this work. Let $\mathcal{F}_k$ be the filtration generated by $\{\theta_0, \cdots, \theta_k, z_0, \cdots, z_k, x_1, \cdots, x_k\}$. For any mapping $g : \mathbb{R}^d \to \mathbb{R}^d$ define the following norm with respect to a function $\mathcal{V} : \mathbb{R}^d \to [1, \infty)$: $\|g\|_{\mathcal{V}} = \sup_{x \in \mathscr{X}} (\|g(x)\|_2 / \mathcal{V}(x))$, and let $L_{\mathcal{V}} = \{g : \mathbb{R}^d \to \mathbb{R}^d, \sup_{x \in \mathscr{X}} \|g\|_{\mathcal{V}} < \infty\}$.

**Assumption 2.1 (Constraint set)** *The set $\Theta \subset \mathbb{R}^d$ is convex and closed with $\max_{x,y \in \Theta} \|x - y\|_2 \leq D_\Theta$, form some $D_\Theta > 0$.*

**Assumption 2.2** *Let $f$ be a continuously differentiable function.*

**Assumption 2.3** *Let $\xi_{k+1}(\theta_k, x_{k+1}) := \nabla F(\theta_k, x_{k+1}) - \nabla f(\theta_k)$. Then,*

$$\mathbb{E}\left[\|\xi_{k+1}(\theta_k, x_{k+1})\|_2^2 \,|\mathcal{F}_k\right] \leq \sigma_1^2 \qquad \mathbb{E}\left[\|\nabla F(\theta_k, x_{k+1})\|_2^2 \,|\mathcal{F}_k\right] \leq \sigma_2^2 \qquad \sigma^2 := \max(\sigma_1^2, \sigma_2^2).$$

**Assumption 2.4** *Let $\{x_k\}_k$ be a Markov chain with transition kernel $P_\theta$. For any $\theta \in \Theta$, $P_\theta$ is irreducible and aperiodic. Additionally, there exists a function $\mathcal{V} : \mathbb{R}^d \to [1, \infty)$ and a constant $\alpha \geq 2$ such that for any compact set $\Theta' \subset \Theta$:*

(a) *There exist a set $C \subset \mathbb{R}^d$, an integer $I$, constants $0 < \lambda < 1$, $b$, $\kappa$, $\delta > 0$, and a probability measure $\nu$ such that,*

$$\sup_{\theta \in \Theta'} P_\theta^l \mathcal{V}^\alpha(x) \leq \lambda \mathcal{V}^\alpha(x) + bI(x \in C) \quad \forall x \in \mathbb{R}^d, \tag{9}$$

$$\sup_{\theta \in \Theta'} P_\theta \mathcal{V}^\alpha(x) \leq \kappa \mathcal{V}^\alpha(x) \quad \forall x \in \mathbb{R}^d, \tag{10}$$

$$\inf_{\theta \in \Theta'} P_\theta^l(x, A) \geq \delta\nu(A) \quad \forall x \in C, \forall A \in \mathcal{B}_{\mathbb{R}^d}. \tag{11}$$

*where $\mathcal{B}_{\mathbb{R}^d}$ is the Borel $\sigma$-algebra over $\mathbb{R}^d$.*

(b) *There exists a constant $c > 0$, such that, for all $x \in \mathbb{R}^d$ and for all $\theta, \theta' \in \Theta'$,*

$$\sup_{\theta \in \Theta'} \|\nabla F(\theta, x)\|_{\mathcal{V}} \leq c, \tag{12}$$

$$\|\nabla F(\theta, x) - \nabla F(\theta', x)\|_{\mathcal{V}} \leq c \|\theta - \theta'\|_2. \tag{13}$$

(c) *There exists a constant $c > 0$, such that, for all $(\theta, \theta') \in \Theta' \times \Theta'$,*

$$\|P_\theta g - P_{\theta'} g\|_{\mathcal{V}} \leq c \|g\|_{\mathcal{V}} \|\theta - \theta'\|_2 \quad \forall g \in L_{\mathcal{V}} \tag{14}$$

$$\|P_\theta g - P_{\theta'} g\|_{\mathcal{V}^\alpha} \leq c \|g\|_{\mathcal{V}^\alpha} \|\theta - \theta'\|_2 \quad \forall g \in L_{\mathcal{V}^\alpha}. \tag{15}$$

Some comments regarding the assumptions are in order. Assumption 2.1, and Assumption 2.2 are common for constrained optimization [GRW20, XBG22, AL22, ZSM$^+$20]. Assumption 2.1, and Assumption 2.2 together imply the Lipschitz continuity of $f(\cdot)$, i.e., there is a constant $L > 0$ such that for any $\theta_1, \theta_2 \in \Theta$, we have $|f(\theta_1) - f(\theta_2)| \leq L \|\theta_1 - \theta_2\|_2$. Assumption 2.3 is common in stochastic optimization literature. Assumption 2.4(a) is a frequently used assumption in Markov chain literature. It implies that for every $\theta \in \Theta$, there exists a stationary distribution $\pi_\theta(x)$, and the chain is $\mathcal{V}^\alpha$-uniformly ergodic [AMP05]. Assumption 2.4(c) provides smoothness guarantee on the function $f(\cdot)$. More formally, we have the following proposition.

**Proposition 2.1 (Lipschitz continuous gradient [AMP05])** *Let Assumption 2.4 be true. Then $f(\cdot)$ has Lipschitz continuous gradient, i.e., there is a constant $L_G > 0$ such that for any $\theta_1, \theta_2 \in \Theta$:*

$$\|\nabla f(\theta_1) - \nabla f(\theta_2)\|_2 \leq L_G \|\theta_1 - \theta_2\|_2. \tag{16}$$

Finally, the most important implication of Assumption 2.4 is that it ensures the existence and regularity of a solution $u(\theta, x)$ to Poisson equation of the transition kernel $P_\theta$ given by $u(\theta, x) - P_\theta u(\theta, x) = \nabla F(\theta, x) - \nabla f(\theta)$. Solution of Poisson equation has been crucial in analyzing additive functionals of Markov chain (see [AMP05] for details). In this work, the Poisson equation solution facilitates a decomposition of the noise as presented in Lemma 3.1 which is a key component of our analysis.

# 3 Main Result

In this section we present our main result on the oracle complexity to establish a bound on $\mathbb{E}[V(\theta_k, z_k)]$. In order to do so we use Algorithm 1, and 2 similar to [XBG22]. If an exact

---

**Algorithm 1** Inexact Averaged Stochastic Approximation (I-ASA)

**Input:** $z_0, \theta_0 \in \mathbb{R}^d$, $\eta_k = (N + k)^{-a}$, $1/2 < a < 1$, $\beta$.
    **for** $k = 1, 2, \cdots, N$ **do**
$$y_k = \begin{cases} \min\limits_{y \in \Theta} \left\{ \langle z_k, y - \theta_k \rangle + \frac{\beta}{2} \|y - \theta_k\|_2^2 \right\} & \text{(Projection)} \\ \text{ICG}(z_k, \theta_k, \beta, t_k, \omega) & \text{(No Projection)} \end{cases}$$
    $\theta_{k+1} = \theta_k + \eta_{k+1}(y_k - \theta_k)$
    $z_{k+1} = (1 - \eta_{k+1})z_k + \eta_{k+1}\nabla F(\theta_k, x_{k+1})$
    **end for**
**Output:** $\theta_R$ where $P(R = i) = \frac{\eta_i}{\sum_{j=1}^N \eta_j}$ for $i = 1, 2, \cdots, N$.

---

**Algorithm 2** Inexact Conditional Gradient (ICG)

**Input:** $z, \theta, \beta, t, \omega$.
    **Set** $w_0 = \theta$
    **for** $i = 1, 2, \cdots, t - 1$ **do**
    Find $v_i$ such that
$$\langle v_i, z + \beta(w_i - \theta) \rangle \leq \operatorname*{argmin}_{v \in \Theta} \langle v, z + \beta(w_i - \theta) \rangle + \beta\omega \mathcal{D}_\Theta^2/(i + 2)$$
    $w_{i+1} = (1 - \mu_i)w_i + \mu_i v_i$ where $\mu_i = \frac{2}{i+2}$
    **end for**
**Output:** $w_t$

---

minimizer of the following subproblem, which is the projection of $\theta_k - z_k/\beta$ on to $\Theta$, is available, then Algorithm 1 is same as ASA algorithm introduced in [GRW20].

$$\min_{y \in \Theta} \left\{ \langle z_k, y - \theta_k \rangle + \frac{\beta}{2} \|y - \theta_k\|_2^2 \right\}. \tag{17}$$

When a projection operator is unavailable or computationally costly, we use Algorithm 2 instead to solve (17). At iteration $k$, Algorithm 2 finds an approximate solution to (17) based on the conditional gradient algorithm. Algorithm 2 needs access to LMO which is often much cheaper and simpler to compute than projection operator. We should emphasize that our results are not limited to ICG method but are valid for any method which can solve (17) within an error of the order of $\{\eta_k\}$.

**Theorem 3.1** *Let Assumption 2.1-2.4 be true. Then, for Algorithm 1,*

    *(a) when a projection operator is available, choosing*

$$\eta_k = (N + k)^{-3/5}, \quad \beta = 1 \tag{18}$$

    *for $k = 1, 2, \cdots, N$ we have*

$$\mathbb{E}[V(\theta_R, z_R)] = \mathcal{O}\left(N^{-\frac{2}{5}}\right),$$

    *(b) when Algorithm 2 is used to solve (17), choosing*

$$\eta_k = (N + k)^{-3/5}, \quad t_k = \eta_k^{-2}, \quad \beta = 1, \quad \omega = 1, \quad \mu_i = 2/(i + 2) \tag{19}$$

*for $k = 1, 2, \cdots, N$ we have*

$$\mathbb{E}\left[V(\theta_R, z_R)\right] = \mathcal{O}\left(N^{-\frac{2}{5}}\right),$$

*where the expectations are taken with respect to all the randomness of the algorithm, and an independent integer random variable $R \in \{1, 2, \cdots, N\}$ with probability mass function,*

$$P\left(R = k\right) = \eta_k / \sum_{k=1}^{N} \eta_k \quad k \in \{1, 2, \cdots, N\}.$$

**Remark 1** *Note that total number of LMO calls are $\sum_{k=1}^{N} t_k = \sum_{k=1}^{N} t_k = \sum_{k=1}^{N} (N+k)^{2a} = \mathcal{O}(N^{11/5})$. In other words, to achieve $\|\mathcal{G}_\Theta(\theta_R, \nabla f(\theta_R), \beta)\|_2^2 \leq \mathbb{E}\left[V(\theta_R, z_R)\right] \leq \epsilon$, SFO and LMO complexities are respectively $\epsilon^{-2.5}$, and $\epsilon^{-5.5}$. Note that the SFO complexity will be $\epsilon^{-2.5}$ as long as one has an approximation of the projection operator with approximation error $\mathcal{O}(\eta_k)$.*

**Remark 2** *In Theorem 3.1, one obtains sublinear rate $\max(N^{a-1}, N^{2-4a})$ with $\eta_k = (N+k)^{-a}$ for $1/2 < a < 1$. Choosing $a = 3/5$ provides the fastest rate of convergence.*

Before sketching the outline of the proof, we present the following lemma which provides a decomposition of the noise $\xi_k(\theta_{k-1}, x_k)$ – one of the key result used in the proof of the main theorem. The lemma and its proof are almost same as Lemma A.5 in [Lia10] with the only difference that unlike [Lia10], where the iterates are of SGD, we need to prove it for the iterates of Algorithm 1. We provide the proof in Appendix A.

**Lemma 3.1** *Let Assumption 2.1-2.4 be true. Then the following decomposition takes place:*
$$\xi_k(\theta_{k-1}, x_k) = e_k + \nu_k + \zeta_k,$$
*where, $\{e_k\}$ is martingale difference sequence, $\mathbb{E}\left[\|\nu_k\|_2\right] \leq \eta_k$, and $\zeta_k = (\tilde{\zeta}_k - \tilde{\zeta}_{k+1})/\eta_k$, where $\mathbb{E}\left[\|\tilde{\zeta}_k\|_2\right] \leq \eta_k$.*

**Outline of the proof of Theorem 3.1:** A key step in the analysis of Algorithm 1 involves controlling the expectation of interaction with noise of the form $\langle \nabla f(\theta_k) - \nabla f(\theta_{k-1}), \xi_{k+1}(\theta_k, x_{k+1}) \rangle$. For iid or martingale difference data it is easy to control because $\mathbb{E}\left[\langle \nabla f(\theta_k) - \nabla f(\theta_{k-1}), \xi_{k+1}(\theta_k, x_{k+1}) \rangle | \mathcal{F}_k\right] = 0$. But this is no longer true for Markov chain data. To resolve the issue, first notice that under our assumptions, the noise sequence $\xi_k$ can be decomposed into the sum of a martingale difference sequence $\{e_k\}$ and some residual terms $\{\nu_k\}$, and $\{\zeta_k\}$ as shown in Lemma 3.1. Then the key step is to introduce a different sequence of hypothetical iterates $(\tilde{\theta}_k, \tilde{y}_k, \tilde{z}_k)$ for which the noise is small enough so that we can bound $\mathbb{E}\left[V(\tilde{\theta}_k, \tilde{z}_k)\right]$, and then show that these hypothetical iterates and the original sequence generated by Algorithm 1 are close enough so that $\mathbb{E}\left[V(\theta_k, z_k)\right]$ is of the same order as $\mathbb{E}\left[V(\tilde{\theta}_k, \tilde{z}_k)\right]$. This step is the main novelty of the proof.

Specifically, consider the following sequence:

$$\tilde{\theta}_0 = \theta_0 \quad \tilde{z}_0 = z_0 \tag{20}$$

$$\tilde{y}_k = \operatorname*{argmin}_{y \in \Theta} \left\{ \left\langle \tilde{z}_k, y - \tilde{\theta}_k \right\rangle + \frac{\beta}{2} \left\| y - \tilde{\theta}_k \right\|_2^2 \right\} \tag{21}$$

$$\tilde{\theta}_{k+1} = \tilde{\theta}_k + \eta_{k+1}(\tilde{y}_k - \tilde{\theta}_k) \tag{22}$$

$$\tilde{z}_{k+1} = z_{k+1} + \tilde{\zeta}_{k+2} \tag{23}$$

This also means,

$$\tilde{z}_{k+1} = (1 - a\eta_{k+1})\tilde{z}_k + a\eta_{k+1}\left(\nabla f(\theta_k) + \tilde{\epsilon}_{k+1}\right), \tag{24}$$

where, $\tilde{\epsilon}_k = e_k + \nu_k + \tilde{\zeta}_k$. Note that by Lemma 3.1, $\mathbb{E}\left[e_k\right] = 0$, and $\mathbb{E}\left[\left\|\nu_k + \tilde{\zeta}_k\right\|_2\right] \leq \eta_k$. First we show that by choosing $\eta_k = (N+k)^{-a}$, $1/2 < a < 1$, and $t_k = 1/\eta_k^2$ one has $\mathbb{E}\left[\|\tilde{\theta}_k - \theta_k\|_2^2\right] = \mathcal{O}\left(N^{2-4a}\right)$, and $\mathbb{E}\left[V(\theta_k, z_k)\right] \leq 2\mathbb{E}\left[V(\tilde{\theta}_k, \tilde{z}_k)\right] + \mathcal{O}\left(N^{2-4a}\right)$. Then we establish the bound on $V(\tilde{\theta}_k, \tilde{z}_k)$. Combining the above two facts proves Theorem 3.1. We defer the detailed proof to Appendix A.1.

## 3.1 State-independent Markov Chain

While our main goal in this work is to analyze Algorithm 1 for constrained nonconvex optimization with state-dependent Markov chain data, we provide the following result on the complexity of Algorithm 1 for Markov chain data with state-independent transition kernel for the sake of completion. Here we use $P$ to denote the transition kernel (as opposed to $P_\theta$ for state-dependent kernel). Note that under Assumption 2.4(a), for each $\theta$, the chain is $\mathcal{V}$-uniformly ergodic, and hence, exponentially mixing [MT12] in the following sense:

**Definition 3** *A Markov chain is said to be exponentially mixing, if there exists $C, r > 0$ such that, for any initial state $x$,*

$$\|P^n(x, \cdot) - \pi\|_\mathcal{V} \leq C \exp(-rn), \tag{25}$$

*where $P^n(x, \cdot)$ is the distribution of $X_n$ with initial state $X_0 = x$.*

Now we present our result on the complexity of Algorithm 1 to find an $\epsilon$-stationary solution to (1) for exponentially-mixing Markov chain data with state-independent transition kernel.

**Theorem 3.2** *Let Assumption 2.1-2.3 be true. Let Assumption 2.4(a)-(b) be true with $P_\theta$ replaced by $P$. Then, for Algorithm 1,*

*(a) when the projection operator is available, choosing*

$$\eta_k = 1/\sqrt{N}, \quad \beta = 1 \tag{26}$$

*for $k = 1, 2, \cdots, N$ we have*

$$\mathbb{E}\left[V(\theta_R, z_R)\right] = \mathcal{O}\left(\log N/\sqrt{N}\right),$$

*(b) when Algorithm 2 is used, choosing*

$$\eta_k = 1/\sqrt{N}, \quad t_k = \lceil\sqrt{k}\rceil, \quad \beta = 1, \quad \omega = 1, \quad \mu_i = 2/(i+2) \tag{27}$$

*for $k = 1, 2, \cdots, N$ we have*

$$\mathbb{E}\left[V(\theta_R, z_R)\right] = \mathcal{O}\left(\log N/\sqrt{N}\right),$$

*where the expectation is taken with respect to all the randomness of the algorithm, and an independent integer random variable $R \in \{1, 2, \cdots, N\}$ whose probability mass function is given by,*

$$P\left(R = k\right) = \eta_k / \sum_{k=1}^{N} \eta_k \quad k \in \{1, 2, \cdots, N\}.$$

We defer the proof to the Appendix.

**Remark 3** *To find an $\epsilon$-stationary point, the total number of calls to SFO and LMO are $\tilde{\mathcal{O}}\left(\epsilon^{-2}\right)$, and $\tilde{\mathcal{O}}\left(\epsilon^{-3}\right)$, where $\tilde{\mathcal{O}}(\cdot)$ denotes the order ignoring logarithmic factors.*

**Remark 4** *The authors of [AL22] obtain the same rate as in Theorem 3.2 for constrained (but projection-based) nonconvex optimization with state-independent exponentially mixing data. In the state-dependent case, since the transition kernel of the Markov chain is controlled by $\theta_k$, and the transition kernel is assumed to be only Lipschitz smooth in $\theta$ (15), the chain does not necessarily exponentially mix. In the state-independent case, since the chain mixes exponentially we obtain the same rate as well. While their results are for projection-based algorithms, we analyze a projection-free LMO-based algorithm since LMO is often computationally cheaper than projection.*

# 4 Experimental Evaluation

## 4.1 Strategic Classification

In this section we illustrate our algorithm on the strategic classification problem as described in Section 1.1 with the GiveMeSomeCredit[4] dataset. The main task is a credit score classification

---

[4]Available at https://www.kaggle.com/c/GiveMeSomeCredit/data

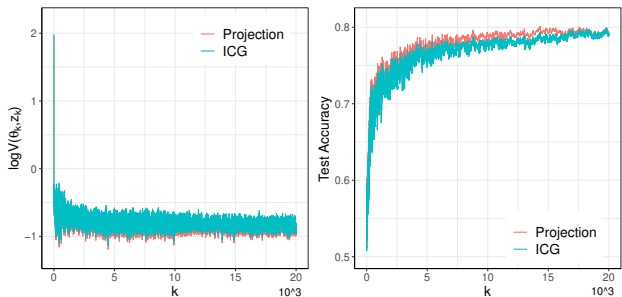

Figure 1: Strategic Classification: (*Left*): Performance of Algorithm 1 with and without the projection operator. (*Right*): Test Accuracy with Algorithm 1 with and without the projection operator.

problem where the bank (learner) has to decide whether a loan should be granted to a client. Given the knowledge of the classifier the clients (agents) can distort some of their personal traits in order to get approved for a loan. Here we use a 2-layer neural network with width $m$ as the classifier, given by

$$h(x; \mathcal{W}, \mathcal{A}, \mathcal{B}) = \sum_{i=1}^{m} \mathcal{A}_i \upsilon(\mathcal{W}_i^\top x + \mathcal{B}_i),$$

where $\upsilon(\cdot)$ is the activation function, $\mathcal{W}_i \in \mathbb{R}^d$, $\mathcal{W} = [\mathcal{W}_1, \mathcal{W}_2, \cdots, \mathcal{W}_d]^\top \in \mathbb{R}^{m \times d}$, $\mathcal{A} = (\mathcal{A}_1, \mathcal{A}_2, \cdots, \mathcal{A}_m) \in \mathbb{R}^m$, $\mathcal{B} = (\mathcal{B}_1, \mathcal{B}_2, \cdots, \mathcal{B}_m) \in \mathbb{R}^m$. We will use $\theta$ to collectively denote $(\mathcal{W}, \mathcal{A}, \mathcal{B})$. We impose the constraint of sparsity on the classifier given by $\|\theta\|_1 \leq R$ for some $R > 0$. As loss function we consider logistic loss as shown in (2). We consider a quadratic cost given by $c(x, x') = \|x_S - x'_S\|_2^2 / (2\lambda)$ where $\lambda$ is the sensitivity of the underlying distribution on $\theta$. We assume that the agents iteratively learn $x'_S$ similar to [LW22]. Note that unlike [LW22], the closed form of best response is not known here. So we assume that the agents use Gradient Ascent (GA) to learn the best response. For $\|\theta\|_1 \leq R$ constraint, the LMO in Algorithm 2 at iteration $k$ is given by $-R \operatorname{sign}(q_i)$, where $i = \operatorname{argmax}_{j=1,\cdots,d} |q_j|$, $q = z + \beta(w_k - \theta)$, and $q_j$ is the $j$-th coordinate of $q$. We select a subset of randomly chosen $M = 2000$ samples (agents) such that the dataset is balanced. Each agent has 10 features. Note that since Algorithm 1 computes the gradient on one sample at every iterate, the computation time is independent of the total number of agents. We assume that the agents can modify Revolving Utilization, Number of Open Credit Lines, and Number of Real Estate Loans or Lines. In this experiment we set $n_1 = 200$. Similar to [LW22], we set $\alpha = 0.5\lambda$, and $\lambda = 0.01$. For the classifier, the activation function is chosen as *sigmoidal*, and $m = 400$. We set $N = 20000$, and $R = 4000$. All the parameters of Algorithm 1 are chosen as described in (19). Figure 4.1 shows that Algorithm 1 finds an $\epsilon$-stationary point of the strategic classification problem. We show that Algorithm 1 performs comparably with Averaged Stochastic Approximation with the projection operator. Each curve in Figure 4.1 is an average of 50 repetitions.

## 4.2 Single Index Model with Trace-norm Ball Constraint

In this section we illustrate our algorithm on a synthetic example of single-index model regression with a nuclear-norm constraint on the model parameter. Let $\|\cdot\|_*$ denote the nuclear norm. The features $\{x_k\}_k \in \mathbb{R}^{d_1 \times d_2}$ are a matrix-valued time-series given by,

$$x_k = Ax_{k-1} + E_k + W_k \upsilon \theta_k,$$

where $A \in \mathbb{R}^{d_1 \times d_1}$ matrix with spectral radius less than 1, $E_k \in \mathbb{R}^{d_1 \times d_2}$ is the noise matrix with each entry of $E_k$ is `iid` $N(0, 1)$ random variable, $W_k$ is a $Bernoulli(0.5)$ random variable, and $\upsilon \in \mathbb{R}$. For a fixed $\theta_k = \theta$, $\{x_k\}_k$ has a stationary distribution as shown in Proposition 1 of [CXY21]. $\{E_k\}_k$, and $\{W_k\}_k$ are `iid` sequence. This Markov chain follows conditions (b) and (c) of Assumption 2.4 since the evolution of $x_k$ only involves linear terms in $\theta_k$. The responses $\{y_k\}_k$ are generated according to the following single index model,

$$y_k = g(x_k^\top \theta^*) + \tilde{E}_k,$$

where $\{\tilde{E}_k\}_k$ is an `iid` sequence of standard normal random variables, $\theta^* \in \mathbb{R}^{d_1 \times d_2}$ is a matrix with $\|\theta^*\|_* \leq 1$, and $g(\cdot) : \mathbb{R} \to \mathbb{R}$ is the link function. For this experiment we choose $g(x) = 3x +$

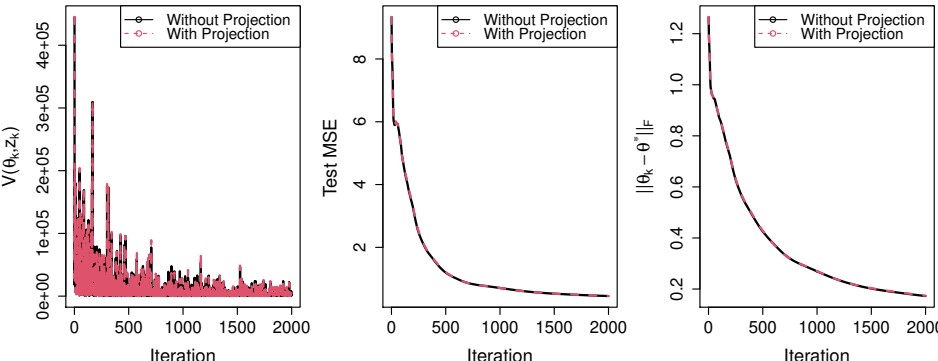

Figure 2: Single-index model with nuclear-norm constraint: (*Left*): Performance of Algorithm 1 with and without the projection operator. (*Middle*): Test Mean Squared Error (MSE) with Algorithm 1 with and without the projection operator. (*Right*): $\|\theta - \theta^*\|_F$ with Algorithm 1 with and without the projection operator.

$5\sin(x)$. Since $y_k$ only depends on $x_k$, and $g$ is a Lipschitz continuous function of $\theta$, Assumption 2.4 holds for $(x_k, y_k)$. It is easy to see that Assumptions 2.1 - 2.3 holds for this example. The constraint set is given by $\|\theta\|_* \le 1$, i.e., we assume that $\theta^*$ has a low-rank structure. The goal is to minimize the expected squared loss with the constraint $\|\theta\|_* \le 1$, i.e.,

$$\min_{\|\theta\|_* \le 1} \mathbb{E}\left[ (y - g(x^\top \theta))^2 \right]. \tag{28}$$

The advantages of conditional-gradient based method for nuclear-norm ball constrained problems have been studied extensively [JS10, Jag13, HJN15]. The main advantage of ICG-based method is that calculating the LMO in this case requires computation of the leading singular vector of gradient matrix whereas to calculate the projection on the trace-norm ball one needs to compute the complete singular value decomposition. Let $u_1$, $v_1$ are the leading left and right singular vectors of the noisy gradient matrix evaluated at $(\theta; x, y)$, $-2(y - g(x^\top \theta))g'(x^\top \theta)x$. Then the LMO is given by $-u_1 v_1^\top$.

For this experiment we choose $d_1 = 10$, $d_2 = 20$, $v = 0.1$, and $N = 2000$. Rest of the parameters of Algorithm 1 are chosen according to Theorem 3.1. In Figure 2, we compare the projection-based and ICG based version of Algorithm 1 with respect to $V(\theta_k, z_k)$, test Mean Squared Error (MSE), and $\|\theta_k - \theta^*\|_F$ where $\|\cdot\|_F$ is the Fröbenius norm. Figure 2 shows that the performance of projection-based and the ICG-based versions of Algorithm 1 are almost same. Each plot in Figure 2 is the average of 50 repetitions.

## 5    Discussion

In this work we provide oracle complexity results for the stochastic conditional gradient algorithm to find an $\epsilon$-stationary point of a constrained nonconvex optimization problem with state-dependent Markovian data. In Theorem 3.1, we show that the number of calls to the SFO and LMO required by the stochastic conditional gradient-type method in Algorithm 1, with *state-dependent* Markovian data, is $\mathcal{O}(\epsilon^{-2.5})$ and $\mathcal{O}(\epsilon^{-5.5})$ respectively. To the best of our knowledge, these are the first oracle complexity results in this setting. In Theorem 3.2, we show that SFO and LMO complexity in the case of state-independent Markovian data is $\tilde{\mathcal{O}}(\epsilon^{-2})$ and $\tilde{\mathcal{O}}(\epsilon^{-3})$ respectively, which matches the corresponding results in the `iid` setting.

There are various avenues for further extensions. Establishing lower bounds on the oracle complexity of projection-free algorithms in the Markovian data setting is extremely interesting. It is also intriguing to establish upper and lower bounds on the oracle complexity for more general types of dependent data sequences arising in applications, including $\phi$ and $\alpha$ mixing sequences. Yet another exciting direction is that of designing algorithms adaptive to the dependency in the data that achieve potentially better oracle complexity bounds.

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
