# A   Appendix

*Proof.* [Proof of Proposition 1.2] Using properties of projection onto a convex set, we have

$$\|\mathcal{G}_\Theta(\theta, z, \beta)\|_2^2 \le 2\beta^2 \|\theta - \Pi_\Theta(\theta - z/\beta)\|_2^2 + 2\beta^2 \|\Pi_\Theta(\theta - z/\beta) - \Pi_\Theta(\theta - \nabla f(\theta)/\beta)\|_2^2$$
$$\le \max(2, 2\beta^2)V(\theta, z).$$

∎

Before proving Lemma 3.1, we present the following result on Poisson equation solution from [Lia10], and [AMP05] which is crucial to the proof of Lemma 3.1.

**Lemma A.1 ([Lia10])** *Let Assumption 2.4 be true. Then we have the following:*

(a) *For any $\theta \in \Theta$, the Markov kernel $P_\theta$ has a single stationary distribution $\pi_\theta$. Moreover, $\nabla F(\theta, x) : \Theta \times \mathbb{R}^d \to \Theta$ is measurable for all $\theta \in \Theta$, $\mathbb{E}_{x \sim \pi_\theta}[\nabla F(\theta, x)] < \infty$.*

(b) *For any $\theta \in \Theta$, the Poisson equation $u(\theta, x) - P_\theta u(\theta, x) = \nabla F(\theta, x) - \nabla f(\theta)$ has a solution $u(\theta, x)$, where $P_\theta u(\theta, x) = \int_{\mathbb{R}^d} u(\theta, x') P_\theta(x, x') dx'$. There exist a function $\mathcal{V} : \mathbb{R}^d \to [1, \infty)$ such that for all $\theta \in \Theta$, the following holds:*

(i) $\sup_{\theta \in \Theta} \|\nabla F(\theta, x)\|_\mathcal{V} < \infty$,
(ii) $\sup_{\theta \in \Theta} (\|u(\theta, x)\|_\mathcal{V} + \|P_\theta u(\theta, x)\|_\mathcal{V}) < \infty$,
(iii) $\sup_{\theta \in \Theta} (\|u(\theta, x) - u(\theta', x)\|_\mathcal{V} + \|P_\theta u(\theta, x) - P_{\theta'} u(\theta', x)\|_\mathcal{V}) < \|\theta - \theta'\|_2$.

We now prove Lemma 3.1.

*Proof.* [Proof of Lemma 3.1] Let,

$$e_{k+1} = u(\theta_k, x_{k+1}) - P_{\theta_k} u(\theta_k, x_k)$$

$$\nu_{k+1} = P_{\theta_{k+1}} u(\theta_{k+1}, x_{k+1}) - P_{\theta_k} u(\theta_k, x_{k+1}) + \frac{\eta_{k+2} - \eta_{k+1}}{\eta_{k+1}} P_{\theta_{k+1}} u(\theta_{k+1}, x_{k+1})$$

$$\tilde{\zeta}_{k+1} = \eta_{k+1} P_{\theta_k} u(\theta_k, x_k)$$

$$\zeta_{k+1} = \frac{\tilde{\zeta}_{k+1} - \tilde{\zeta}_{k+2}}{\eta_{k+1}}. \tag{29}$$

Now, one has,

$$\mathbb{E}[e_{k+1}|\mathcal{F}_k] = \mathbb{E}[u(\theta_k, x_{k+1})|\mathcal{F}_k] - P_{\theta_k} u(\theta_k, x_k) = 0$$

We also have $\mathbb{E}[|e_{k+1}|] < \infty$. So $e_{k+1}$ is a martingale difference sequence. We also have, using Lemma A.1, and the fact that $\Theta$ is compact,

$$\mathbb{E}[\|\nu_{k+1}\|_2] \le c_1 \|\theta_k - \theta_{k+1}\|_2 + c_2 \eta_{k+2} \le c_1 \eta_{k+1} \|y_k - \theta_k\|_2 + c_2 \eta_{k+2} \le c_3 \eta_{k+1}.$$

Again, using Lemma A.1, we have

$$\mathbb{E}\left[\left\|\tilde{\zeta}_{k+1}\right\|_2\right] \le \eta_{k+1} \mathbb{E}\left[\|P_{\theta_k} u(\theta_k, x_k)\|_2\right] \le c_4 \eta_{k+1},$$

where $c_i$, $i = 1, 2, 3, 4$ are constants.

∎

## A.1   Proof of Theorem 3.1

Let,

$$y_k' = \operatorname*{argmin}_{y \in \Theta} \left\{ \langle z_k, y - \theta_k \rangle + \frac{\beta}{2} \|y - \theta_k\|_2^2 \right\},$$

and,

$$\|y_k - y_k'\|_2 \le \delta_k.$$

Consider the following system:

$$\tilde{\theta}_0 = \theta_0 \quad \tilde{z}_0 = z_0 \tag{30}$$

$$\tilde{y}_k = \operatorname*{argmin}_{y \in \Theta} \left\{ \left\langle \tilde{z}_k, y - \tilde{\theta}_k \right\rangle + \frac{\beta}{2} \left\| y - \tilde{\theta}_k \right\|_2^2 \right\} \tag{31}$$

$$\tilde{\theta}_{k+1} = \tilde{\theta}_k + \eta_{k+1}(\tilde{y}_k - \tilde{\theta}_k) \tag{32}$$

$$\tilde{z}_{k+1} = z_{k+1} + \tilde{\zeta}_{k+1} \tag{33}$$

Equivalently one can also write:

$$\tilde{y}_k = \Pi_\Theta \left( \tilde{\theta}_k - \frac{1}{\beta} \tilde{z}_k \right), \tag{34}$$

where $\Pi_\Theta$ is the orthogonal projection on the set $\Theta$. Let $\phi(\theta, z)$ be the following function:

$$\phi(\theta, z) = \min_{y \in \Theta} \left( \langle z, y - \theta \rangle + \frac{\beta}{2} \|y - \theta\|_2^2 \right). \tag{35}$$

Let us define the following merit function.

$$W(\theta, z) = (f(\theta) - f^*) - \phi(\theta, z). \tag{36}$$

Recall that as optimality measure we use the following:

$$V(\theta_k, z_k) = \left\| \Pi_\Theta \left( \theta_k - \frac{z_k}{\beta} \right) - \theta_k \right\|_2^2 + \|z_k - \nabla f(\theta_k)\|_2^2. \tag{37}$$

**Lemma A.2 ([Jag13])** *Under Assumption 2.1,*

$$\|y_k - y_k'\|_2^2 \le \frac{4\mathcal{D}_\Theta^2(1 + \omega)}{t_k + 2},$$

*where $\omega$ is the accuracy of the LMO.*

In Lemma A.3, we show that the iterates generated by the auxiliary updates are close to the original updates of Algorithm 1. Using Lemma A.3 we show that $V(\theta_k, z_k)$ is close to $V(\tilde{\theta}_k, \tilde{z}_k)$ in Lemma A.4. Lemma A.5, and Lemma A.6 bounds the first component of $\sum_{k=1}^N \eta_k V(\theta_k, z_k)$, namely, $\sum_{k=1}^N \eta_k \left\| \Pi_\Theta \left( \theta_k - \frac{z_k}{\beta} \right) - \theta_k \right\|_2^2$. Finally, we bound the second component $\sum_{k=1}^N \|z_k - \nabla f(\theta_k)\|_2^2$ in (46).

**Lemma A.3** *Let the conditions of Lemma 3.1 hold. Then, for $k \ge 1$, and for any $\gamma \in \mathbb{R}$, we have*

$$\mathbb{E}\left[ \left\| \tilde{\theta}_k - \theta_k \right\|_2^2 \right] \le \eta_k^3 (1 + \eta_k^{-\gamma}) + 2 \sum_{i=1}^{k-1} \eta_i^3 \left( 1 + \eta_i^{-\gamma} \right) \prod_{j=i+1}^k \left( 1 + \eta_j^{1+\gamma} \right)$$

$$+ 2 \sum_{i=1}^{k-1} \eta_i \mathbb{E}\left[ \delta_{i-1}^2 \right] \left( 1 + \eta_i^{-\gamma} \right) \prod_{j=i+1}^k \left( 1 + \eta_j^{1+\gamma} \right). \tag{38}$$

**Lemma A.4** *Let the conditions of Lemma 3.1 be true. Then, choosing $\eta_k = (N + k)^{-a}$, $a > 1/2$, and setting $\gamma$ of Lemma A.3 to $\gamma = 1/a - 1$, for $\delta_k \le \eta_k$ we get*

$$\mathbb{E}\left[ V(\theta_k, z_k) \right] \le 2\mathbb{E}\left[ V(\tilde{\theta}_k, \tilde{z}_k) \right] + (9 + 4L_G) \left( N^{1-4a} + 8N^{2-4a} \right) + 12N^{-2a}.$$

**Lemma A.5** *Let Assumption 2.1, Assumption 2.2, and Assumption 2.4 be true. Let $\{\tilde{\theta}_k, \tilde{z}_k, \tilde{y}_k, \}_{k \ge 0}$ be the sequence generated by (30)-(33). Then $\forall k \ge 0$,*

$$\frac{\beta}{2} \sum_{k=0}^{N-1} \eta_{k+1} \left\| \tilde{y}_k - \tilde{\theta}_k \right\|_2^2 \le W(x_0, z_0) + \sum_{k=0}^{N-1} r_{k+1} \quad \forall N \ge 1, \tag{39}$$

*where for $k \ge 0$,*

$$r_{k+1} = \frac{(L_G + L_\phi)\eta_{k+1}^2}{2} \left\| \tilde{y}_k - \tilde{\theta}_k \right\|_2^2 + \frac{L_\phi}{2} \|\tilde{z}_{k+1} - \tilde{z}_k\|_2^2 \tag{40}$$

$$+ \eta_{k+1} \left\langle \tilde{\theta}_k - \tilde{y}_k, \tilde{\epsilon}_{k+1} \right\rangle + \frac{\eta_{k+1} L_G^2}{\beta} \left\| \theta_k - \tilde{\theta}_k \right\|_2^2.$$

**Lemma A.6** *Let $\{\tilde{\theta}_k, \tilde{z}_k, \tilde{y}_k, \}_{k \geq 0}$ be the sequence generated by* (30)-(33)*, and Assumption 2.1-2.4 hold. Then,*

1. *If $\eta_0 = 1$, we have,*

$$\beta^2 \mathbb{E}\left[\left\|\tilde{y}_k - \tilde{\theta}_k\right\|_2^2 | \mathcal{F}_{k-1}\right] \leq \mathbb{E}\left[\|\tilde{z}_k\|_2^2 | \mathcal{F}_{k-1}\right] \leq \sigma^2 \quad \forall k \geq 1; \tag{41}$$

2. *If $\eta_k \leq 1/\sqrt{2}$ for all $k \geq 1$, then,*

$$\sum_{k=0}^{\infty} \mathbb{E}\left[\|\tilde{z}_{k+1} - \tilde{z}_k\|_2^2 | \mathcal{F}_k\right] \leq 2\left(\|\tilde{z}_0\|_2^2 + 24\sigma^2 \sum_{k=0}^{\infty} \eta_k^2\right), \tag{42}$$

$$\sum_{k=0}^{\infty} \mathbb{E}\left[r_{k+1} | \mathcal{F}_k\right] \leq \sigma_3^2 \sum_{k=0}^{\infty} \eta_k^2 + \frac{L_G^2}{\beta} \sum_{k=0}^{\infty} \eta_{k+1} \mathbb{E}\left[\left\|\theta_k - \tilde{\theta}_k\right\|_2^2\right], \tag{43}$$

*where*

$$\sigma_3^2 = \frac{1}{2}\left((3L_G + L_\phi)\frac{\sigma^2}{\beta^2} + 4L_\phi(\|z_0\|_2^2 + 24\sigma^2) + 2\right).$$

We now prove Theorem 3.1.

*Proof.* [Proof of Theorem 3.1] Define,

$$\Gamma_1 := 1 \qquad \Gamma_k := \prod_{i=0}^{k-1}(1 - \eta_{i+1}) \quad \forall k \geq 2. \tag{44}$$

Now, using the update of Algorithm 1,

$$\nabla f(\tilde{\theta}_{k+1}) - \tilde{z}_{k+1} = (1 - \eta_{k+1})\left(\nabla f(\tilde{\theta}_k) - \tilde{z}_k + \nabla f(\tilde{\theta}_{k+1}) - \nabla f(\tilde{\theta}_k)\right)$$
$$+ \eta_{k+1}\left(\nabla f(\tilde{\theta}_{k+1}) - \nabla f(\tilde{\theta}_k) - \tilde{\epsilon}_{k+1}\right) + \eta_{k+1}\left(\nabla f(\tilde{\theta}_k) - \nabla f(\theta_k)\right)$$

Dividing both sides by $\Gamma_{k+1}$, we get,

$$\frac{\nabla f(\tilde{\theta}_{k+1}) - \tilde{z}_{k+1}}{\Gamma_{k+1}}$$
$$= \frac{1}{\Gamma_k}\left(\nabla f(\tilde{\theta}_k) - \tilde{z}_k + \nabla f(\tilde{\theta}_{k+1}) - \nabla f(\tilde{\theta}_k)\right) + \frac{\eta_{k+1}}{\Gamma_{k+1}}\left(\nabla f(\tilde{\theta}_{k+1}) - \nabla f(\tilde{\theta}_k) - \tilde{\epsilon}_{k+1}\right)$$
$$+ \frac{\eta_{k+1}}{\Gamma_{k+1}}\left(\nabla f(\tilde{\theta}_k) - \nabla f(\theta_k)\right)$$
$$= \frac{1}{\Gamma_k}\left(\nabla f(\tilde{\theta}_k) - \tilde{z}_k\right) + \frac{1}{\Gamma_{k+1}}\left(\nabla f(\tilde{\theta}_{k+1}) - \nabla f(\tilde{\theta}_k)\right) - \frac{\eta_{k+1}}{\Gamma_{k+1}}\left(\tilde{\epsilon}_{k+1} + \nabla f(\theta_k) - \nabla f(\tilde{\theta}_k)\right)$$

Summing both sides from $k = 1$ to $k = i - 1$, we get,

$$\nabla f(\tilde{\theta}_i) - \tilde{z}_i = \sum_{k=0}^{i-1}\frac{\Gamma_i}{\Gamma_{k+1}}\left(\nabla f(\tilde{\theta}_{k+1}) - \nabla f(\tilde{\theta}_k)\right) - \sum_{k=0}^{i-1}\frac{\eta_{k+1}\Gamma_i}{\Gamma_{k+1}}\left(\tilde{\epsilon}_{k+1} + \nabla f(\theta_k) - \nabla f(\tilde{\theta}_k)\right).$$

Then,

$$\nabla f(\tilde{\theta}_i) - \tilde{z}_i = \frac{\Gamma_i}{\Gamma_{i-1}}\left(\nabla f(\tilde{\theta}_{i-1}) - \tilde{z}_{i-1}\right) + \left(\nabla f(\tilde{\theta}_i) - \nabla f(\tilde{\theta}_{i-1})\right) - \eta_i\left(\tilde{\epsilon}_i + \nabla f(\theta_{i-1}) - \nabla f(\tilde{\theta}_{i-1})\right)$$
$$= (1 - \eta_i)\left(\nabla f(\tilde{\theta}_{i-1}) - \tilde{z}_{i-1}\right) + \frac{\eta_i}{\eta_i}\left(\nabla f(\tilde{\theta}_i) - \nabla f(\tilde{\theta}_{i-1})\right) - \eta_i\left(\tilde{\epsilon}_i + \nabla f(\theta_{i-1}) - \nabla f(\tilde{\theta}_{i-1})\right)$$

Using Young's inequality and Jensen's inequality,

$$\left\|\nabla f(\tilde{\theta}_i) - \tilde{z}_i\right\|_2^2$$

$$\leq \frac{1-\eta_i/4}{1-\eta_i/2} \left\| (1-\eta_i)\left(\nabla f(\tilde{\theta}_{i-1}) - \tilde{z}_{i-1}\right) + \frac{\eta_i}{\eta_i}\left(\nabla f(\tilde{\theta}_i) - \nabla f(\tilde{\theta}_{i-1})\right) - \eta_i \tilde{\epsilon}_i \right\|_2^2$$

$$+ \frac{4-\eta_i}{\eta_i}\eta_i^2 \left\| \nabla f(\theta_{i-1}) - \nabla f(\tilde{\theta}_{i-1}) \right\|_2^2$$

$$\leq \frac{1-\eta_i/4}{1-\eta_i/2}(I_1) + 4L_G^2 \eta_i \left\| \theta_{i-1} - \tilde{\theta}_{i-1} \right\|_2^2, \tag{45}$$

where

$$I_1 = (1-\eta_i)\left\| \nabla f(\tilde{\theta}_{i-1}) - \tilde{z}_{i-1} \right\|_2^2 + \frac{\left\| \nabla f(\tilde{\theta}_i) - \nabla f(\tilde{\theta}_{i-1}) \right\|_2^2}{\eta_i} + \eta_i^2 \left\| \tilde{\epsilon}_i \right\|_2^2$$

$$- 2\eta_i \left\langle (1-\eta_i)\left(\nabla f(\tilde{\theta}_{i-1}) - \tilde{z}_{i-1}\right) + \left(\nabla f(\tilde{\theta}_i) - \nabla f(\tilde{\theta}_{i-1})\right), \tilde{\epsilon}_i \right\rangle.$$

Taking conditional expectation of $I_1$ with respect to $\mathcal{F}_{i-1}$, using (16), Assumption 2.4, and (32), we get

$$\mathbb{E}\left[I_1 \mid \mathcal{F}_{i-1}\right] \leq (1-\eta_i)\left\| \nabla f(\tilde{\theta}_{i-1}) - \tilde{z}_{i-1} \right\|_2^2 + \eta_i L_G^2 \left\| \tilde{y}_{i-1} - \tilde{\theta}_{i-1} \right\|_2^2 + \eta_i^2 \sigma^2$$

$$- 2\eta_i \mathbb{E}\left[ \left\langle (1-\eta_i)\left(\nabla f(\tilde{\theta}_{i-1}) - \tilde{z}_{i-1}\right) + \left(\nabla f(\tilde{\theta}_i) - \nabla f(\tilde{\theta}_{i-1})\right), \tilde{\zeta}_i \right\rangle \mid \mathcal{F}_{i-1}\right]$$

$$\leq (1-\eta_i)\left\| \nabla f(\tilde{\theta}_{i-1}) - \tilde{z}_{i-1} \right\|_2^2 + \eta_i L_G^2 \mathbb{E}\left[ \left\| \tilde{y}_{i-1} - \tilde{\theta}_{i-1} \right\|_2^2 \mid \mathcal{F}_{i-1}\right] + \eta_i^2 \sigma^2$$

$$+ 2\eta_i^2 (1-\eta_i)^2 \left\| \nabla f(\tilde{\theta}_{i-1}) - \tilde{z}_{i-1} \right\|_2^2 + 2\eta_i^4 L_G^2 \left\| \tilde{y}_{i-1} - \tilde{\theta}_{i-1} \right\|_2^2 + \eta_i^2$$

$$\leq \left(1 - \frac{\eta_i}{2}\right)\left\| \nabla f(\tilde{\theta}_{i-1}) - \tilde{z}_{i-1} \right\|_2^2 + 2\eta_i L_G^2 \mathbb{E}\left[ \left\| \tilde{y}_{i-1} - \tilde{\theta}_{i-1} \right\|_2^2 \mid \mathcal{F}_{i-1}\right] + \eta_i^2 (1+\sigma^2).$$

Taking expectation on both sides of (45),

$$\mathbb{E}\left[\left\| \nabla f(\tilde{\theta}_i) - \tilde{z}_i \right\|_2^2\right] \leq \left(1 - \frac{\eta_i}{4}\right)\mathbb{E}\left[\left\| \nabla f(\tilde{\theta}_{i-1}) - \tilde{z}_{i-1} \right\|_2^2\right] + 4\eta_i L_G^2 \mathbb{E}\left[\left\| \tilde{y}_{i-1} - \tilde{\theta}_{i-1} \right\|_2^2\right]$$

$$+ 2\eta_i^2(1+\sigma^2) + 4L_G^2 \eta_i \mathbb{E}\left[\left\| \theta_{i-1} - \tilde{\theta}_{i-1} \right\|_2^2\right]$$

$$\leq Y_0^i \mathbb{E}\left[\left\| \nabla f(\tilde{\theta}_0) - \tilde{z}_0 \right\|_2^2\right] + 4L_G^2 \sum_{k=1}^{i} Y_k^i \eta_k \mathbb{E}\left[\left\| \tilde{y}_{k-1} - \tilde{\theta}_{k-1} \right\|_2^2\right] + 2\sum_{k=1}^{i} Y_{k-1}^i \eta_k^2 (1+\sigma^2)$$

$$+ 4L_G^2 \sum_{k=1}^{i} Y_{k-1}^i \eta_k \mathbb{E}\left[\left\| \theta_{k-1} - \tilde{\theta}_{k-1} \right\|_2^2\right],$$

where

$$Y_i^i = \mathbf{I} \qquad Y_i^k = \prod_{j=i+1}^{k}\left(1 - \frac{\eta_j}{4}\right) \ \text{ for } k > i.$$

Then,

$$\sum_{i=1}^{N} \eta_i \mathbb{E}\left[\left\| \nabla f(\tilde{\theta}_i) - \tilde{z}_i \right\|_2^2\right]$$

$$\leq \sum_{i=1}^{N} \eta_i Y_0^i \mathbb{E}\left[\left\| \nabla f(\tilde{\theta}_0) - \tilde{z}_0 \right\|_2^2\right] + 4L_G^2 \sum_{i=1}^{N}\sum_{k=1}^{i} Y_k^i \eta_i \eta_k \mathbb{E}\left[\left\| \tilde{y}_{k-1} - \tilde{\theta}_{k-1} \right\|_2^2\right]$$

$$+ 2\sum_{i=1}^{N}\sum_{k=1}^{i} Y_{k-1}^i \eta_i \eta_k^2 (1+\sigma^2) + 4L_G^2 \sum_{i=1}^{N}\sum_{k=1}^{i} Y_{k-1}^i \eta_i \eta_k \mathbb{E}\left[\left\| \theta_{k-1} - \tilde{\theta}_{k-1} \right\|_2^2\right]$$

$$=\mathbb{E}\left[\left\|\nabla f(\tilde{\theta}_0) - \tilde{z}_0\right\|_2^2\right] + 4L_G^2 \sum_{k=1}^{N}\sum_{i=k}^{N} Y_k^i \eta_i \eta_k \mathbb{E}\left[\left\|\tilde{y}_{k-1} - \tilde{\theta}_{k-1}\right\|_2^2\right]$$

$$+ 2\sum_{k=1}^{N}\sum_{i=k}^{N} Y_{k-1}^i \eta_i \eta_k^2 (1+\sigma^2) + 4L_G^2 \sum_{k=1}^{N}\sum_{i=k}^{N} Y_{k-1}^i \eta_i \eta_k \mathbb{E}\left[\left\|\theta_{k-1} - \tilde{\theta}_{k-1}\right\|_2^2\right]$$

$$\leq \mathbb{E}\left[\left\|\nabla f(\tilde{\theta}_0) - \tilde{z}_0\right\|_2^2\right] + 4L_G^2 \sum_{k=0}^{N-1} \eta_k \mathbb{E}\left[\left\|\tilde{y}_k - \tilde{\theta}_k\right\|_2^2\right] + 2\sum_{k=1}^{N} \eta_k^2 (1+\sigma^2) + 4L_G^2 \sum_{k=0}^{N-1} \eta_{k+1} \mathbb{E}\left[\left\|\theta_k - \tilde{\theta}_k\right\|_2^2\right].$$

The last inequality follows by Lemma A.7. Combining (39), and (43), we get,

$$\sum_{i=1}^{N} \eta_i \mathbb{E}\left[\left\|\nabla f(\tilde{\theta}_i) - \tilde{z}_i\right\|_2^2\right] \leq \mathbb{E}\left[\left\|\nabla f(\tilde{\theta}_0) - \tilde{z}_0\right\|_2^2\right]$$

$$+ 4L_G^2 \left(W(x_0, z_0) + \sigma^2 \sum_{k=0}^{N} \eta_k^2 + \frac{L_G^2}{\beta} \sum_{k=0}^{\infty} \eta_{k+1} \mathbb{E}\left[\left\|\theta_k - \tilde{\theta}_k\right\|_2^2\right]\right)$$

$$+ \sum_{k=1}^{N} \eta_k^2 (1+\sigma^2) + 4L_G^2 \sum_{k=0}^{N-1} \eta_{k+1} \mathbb{E}\left[\left\|\theta_k - \tilde{\theta}_k\right\|_2^2\right]. \tag{46}$$

Combining (46), and (39), we get,

$$\mathbb{E}\left[V(\tilde{\theta}_k, \tilde{z}_k)\right]$$

$$\leq \frac{\mathbb{E}\left[\left\|\nabla f(\tilde{\theta}_0) - \tilde{z}_0\right\|_2^2\right] + 4L_G^2 W(x_0, z_0) + 2\sum_{k=1}^{N} \eta_k^2 (1+\sigma^2) + (4L_G^2 + 4L_G^4/\beta) \sum_{k=0}^{N-1} \eta_{k+1} \mathbb{E}\left[\left\|\theta_k - \tilde{\theta}_k\right\|_2^2\right]}{\sum_{k=1}^{N} \eta_k}.$$
$$\tag{47}$$

Choosing $\eta_k = (N+k)^{-a}$, using Lemma A.3, for $\gamma = 1/a - 1$, we get,

$$\sum_{k=0}^{N-1} \eta_{k+1} \mathbb{E}\left[\left\|\theta_k - \tilde{\theta}_k\right\|_2^2\right]$$

$$\leq \sum_{k=0}^{N-1} \eta_{k+1} \eta_k^3 (1 + \eta_k^{-\gamma}) + \sum_{k=0}^{N-1} \eta_{k+1} \sum_{i=1}^{k-1} \eta_i^3 \left(1 + \eta_i^{-\gamma}\right) \prod_{j=i+1}^{k} \left(1 + \eta_j^{1+\gamma}\right)$$

$$\leq 2\sum_{k=0}^{N-1} (N+k+1)^{-a} \sum_{i=1}^{k-1} (N+i)^{-3a} \left(1 + (N+i)^{a\gamma}\right) \prod_{j=i+1}^{k} \left(1 + (N+j)^{-a(1+\gamma)}\right)$$

$$\leq 2\sum_{k=0}^{N-1} N^{-4a} \sum_{i=1}^{k-1} \left(1 + (2N)^{1-a}\right) \left(1 + N^{-1}\right)^{k-i}$$

$$\leq 2\sum_{k=0}^{N-1} N^{1-4a} \left(1 + (2N)^{1-a}\right) \left((1 + N^{-1})^k - 1\right)$$

$$\leq 8N^{2-5a} \left(N(1 + N^{-1})^N - 2N\right)$$

$$\leq 8N^{3-5a}.$$

Then,

$$\frac{\sum_{k=0}^{N-1} \eta_{k+1} \mathbb{E}\left[\left\|\theta_k - \tilde{\theta}_k\right\|_2^2\right]}{\sum_{k=1}^{N} \eta_k} \leq \frac{8N^{3-5a}}{\sum_{k=1}^{N} (2N)^{-a}} \leq 16N^{2-4a}. \tag{48}$$

Now using (48), and (47), we get

$$\mathbb{E}\left[V(\tilde{\theta}_k, \tilde{z}_k)\right] \leq \left(\mathbb{E}\left[\left\|\nabla f(\tilde{\theta}_0) - \tilde{z}_0\right\|_2^2\right] + 4L_G^2 W(x_0, z_0)\right) N^{a-1} + 2(1+\sigma^2)N^{-a} + 16N^{2-4a}.$$

Then using Lemma A.4, we get,

$$
\begin{aligned}
\mathbb{E}\left[V(\theta_k, z_k)\right] \leq & \left(\mathbb{E}\left[\left\|\nabla f(\tilde{\theta}_0) - \tilde{z}_0\right\|_2^2\right] + 4L_G^2 W(x_0, z_0)\right) N^{a-1} \\
& + 2(1+\sigma^2)N^{-a} + (9 + 4L_G)\left(N^{1-4a} + 8N^{2-4a}\right) + 12N^{-2a} + 16N^{2-4a}.
\end{aligned}
$$
(49)

Now choosing, $a = 3/5$, we get,

$$
\begin{aligned}
\mathbb{E}\left[V(\theta_k, z_k)\right] \leq & \left(\mathbb{E}\left[\left\|\nabla f(\tilde{\theta}_0) - \tilde{z}_0\right\|_2^2\right] + 4L_G^2 W(x_0, z_0)\right) N^{-2/5} \\
& + 2(1+\sigma^2)N^{-3/5} + (9 + 4L_G)\left(N^{-7/5} + 8N^{-2/5}\right) + 12N^{-6/5} + 16N^{-2/5} \\
= & \mathcal{O}\left(N^{-\frac{2}{5}}\right).
\end{aligned}
$$

∎

Now we the provide the proofs of the Lemmas required to prove Theorem 3.1.

## A.2 Proof of Lemmas for Theorem 3.1

*Proof.* [Proof of Lemma A.3] By Jensen's inequality, contraction property of the projection operator, and Youngs' inequality, we get

$$
\begin{aligned}
& \left\|\tilde{\theta}_{k+1} - \theta_{k+1}\right\|_2^2 \\
\leq & (1 - \eta_{k+1})\left\|\tilde{\theta}_k - \theta_k\right\|_2^2 + \eta_{k+1}\left\|\tilde{y}_k - y_k\right\|_2^2 \\
\leq & (1 - \eta_{k+1})\left\|\tilde{\theta}_k - \theta_k\right\|_2^2 + \eta_{k+1}\left(\left\|\tilde{\theta}_k - \theta_k\right\|_2 + \left\|\tilde{z}_k/\beta - z_k/\beta\right\|_2 + \delta_k\right)^2 \\
\leq & (1 - \eta_{k+1})\left\|\tilde{\theta}_k - \theta_k\right\|_2^2 + \eta_{k+1}(1 + \eta_{k+1}^\gamma)\left\|\tilde{\theta}_k - \theta_k\right\|_2^2 + 2\eta_{k+1}(1 + \eta_{k+1}^{-\gamma})\left\|\tilde{z}_k/\beta - z_k/\beta\right\|_2^2 + 2\eta_{k+1}(1 + \eta_{k+1}^{-\gamma})\delta_k^2.
\end{aligned}
$$

Now taking expectation on both sides, and using Lemma 3.1, we have

$$
\begin{aligned}
& \mathbb{E}\left[\left\|\tilde{\theta}_{k+1} - \theta_{k+1}\right\|_2^2\right] \\
\leq & \left(1 + \eta_{k+1}^{1+\gamma}\right)\mathbb{E}\left[\left\|\tilde{\theta}_k - \theta_k\right\|_2^2\right] + 2\eta_{k+1}^3(1 + \eta_{k+1}^{-\gamma}) + 2\eta_{k+1}(1 + \eta_{k+1}^{-\gamma})\mathbb{E}\left[\delta_k^2\right] \\
\leq & \eta_{k+1}^3(1 + \eta_{k+1}^{-\gamma}) + 2\sum_{i=1}^k \eta_i^3\left(1 + \eta_i^{-\gamma}\right)\prod_{j=i+1}^{k+1}\left(1 + \eta_j^{1+\gamma}\right) + 2\sum_{i=1}^k \eta_i \mathbb{E}\left[\delta_{i-1}^2\right]\left(1 + \eta_i^{-\gamma}\right)\prod_{j=i+1}^{k+1}\left(1 + \eta_j^{1+\gamma}\right).
\end{aligned}
$$

∎

*Proof.* [Proof of Lemma A.4] Using (16), and contraction property of the projection operator,

$$
\begin{aligned}
V(\theta_k, z_k) = & \left\|\Pi_\Theta\left(\theta_k - z_k\right) - \theta_k\right\|_2^2 + \left\|z_k - \nabla f(\theta_k)\right\|_2^2 \\
\leq & 2\left\|\Pi_\Theta\left(\theta_k - z_k\right) - \theta_k - \Pi_\Theta\left(\tilde{\theta}_k - \tilde{z}_k\right) + \tilde{\theta}_k\right\|_2^2 + 2\left\|\tilde{z}_k - \nabla f(\tilde{\theta}_k) - z_k + \nabla f(\theta_k)\right\|_2^2 \\
& + 2\left\|\Pi_\Theta\left(\tilde{\theta}_k - \tilde{z}_k\right) - \tilde{\theta}_k\right\|_2^2 + 2\left\|\tilde{z}_k - \nabla f(\tilde{\theta}_k)\right\|_2^2 \\
\leq & 2V(\tilde{\theta}_k, \tilde{z}_k) + (8 + 4L_G)\left\|\theta_k - \tilde{\theta}_k\right\|_2^2 + 12\left\|z_k - \tilde{z}_k\right\|_2^2.
\end{aligned}
$$

Using Lemma A.3, and Lemma 3.1, we get,

$$
\mathbb{E}\left[V(\theta_k, z_k)\right]
$$

$$\leq 2\mathbb{E}\left[V(\tilde{\theta}_k, \tilde{z}_k)\right] + (8 + 4L_G)\mathbb{E}\left[\left\|\theta_k - \tilde{\theta}_k\right\|_2^2\right] + 12\mathbb{E}\left[\|z_k - \tilde{z}_k\|_2^2\right]$$

$$\leq 2\mathbb{E}\left[V(\tilde{\theta}_k, \tilde{z}_k)\right] + (8 + 4L_G)\left(\eta_k^3(1 + \eta_k^{-\gamma}) + 2\sum_{i=1}^{k-1}\eta_i^3\left(1 + \eta_i^{-\gamma}\right)\prod_{j=i+1}^{k}\left(1 + \eta_j^{1+\gamma}\right)\right.$$

$$\left. + 2\sum_{i=1}^{k-1}\eta_i\mathbb{E}\left[\delta_{i-1}^2\right]\left(1 + \eta_i^{-\gamma}\right)\prod_{j=i+1}^{k}\left(1 + \eta_j^{1+\gamma}\right)\right) + 12\eta_{k+1}^2.$$

For $\delta_{k-1} \leq \eta_k$, choosing $\eta_k = \frac{1}{(N+k)^a}$ with $a > 1/2$, we get,

$$\mathbb{E}\left[V(\theta_k, z_k)\right]$$

$$\leq 2\mathbb{E}\left[V(\tilde{\theta}_k, \tilde{z}_k)\right] + (8 + 4L_G)\left(\frac{1 + (N+k)^{a\gamma}}{(N+k)^{3a}} + 4\sum_{i=1}^{k-1}\frac{1 + (N+i)^{a\gamma}}{(N+i)^{3a}}\prod_{j=i+1}^{k}\left(1 + (N+j)^{-a(1+\gamma)}\right)\right)$$

$$+ \frac{12}{(N+k+1)^{2a}}$$

$$\leq 2\mathbb{E}\left[V(\tilde{\theta}_k, \tilde{z}_k)\right] + (9 + 4L_G)\left(\frac{1}{N^{3a-a\gamma}} + \sum_{i=1}^{k-1}\frac{4}{N^{3a-a\gamma}}\left(1 + \frac{1}{N^{a(1+\gamma)}}\right)^i\right) + \frac{12}{N^{2a}}$$

$$\leq 2\mathbb{E}\left[V(\tilde{\theta}_k, \tilde{z}_k)\right] + (9 + 4L_G)\left(\frac{1}{N^{3a-a\gamma}} + \frac{4}{N^{2a-2a\gamma}}\left[\left(1 + \frac{1}{N^{a(1+\gamma)}}\right)^N - 1\right]\right) + \frac{12}{N^{2a}}$$

$$\leq 2\mathbb{E}\left[V(\tilde{\theta}_k, \tilde{z}_k)\right] + (9 + 4L_G)\left(\frac{1}{N^{3a-a\gamma}} + \frac{4}{N^{2a-2a\gamma}}\left[\exp\left(N^{1-a(1+\gamma)}\right) - 1\right]\right) + \frac{12}{N^{2a}}$$

$$\leq 2\mathbb{E}\left[V(\tilde{\theta}_k, \tilde{z}_k)\right] + (9 + 4L_G)\left(\frac{1}{N^{3a-a\gamma}} + 8N^{1-3a+a\gamma}\right) + \frac{12}{N^{2a}}. \tag{50}$$

Setting $\gamma = 1/a - 1$, we get,

$$\mathbb{E}\left[V(\theta_k, z_k)\right] \leq 2\mathbb{E}\left[V(\tilde{\theta}_k, \tilde{z}_k)\right] + (9 + 4L_G)\left(N^{1-4a} + 8N^{2-4a}\right) + 12N^{-2a}.$$

∎

*Proof.* [Proof of Lemma A.5]

Recall that,

$$\phi(\theta, z) = \min_{y \in \Theta}\left(\langle z, y - \theta\rangle + \frac{\beta}{2}\|y - \theta\|_2^2\right). \tag{51}$$

It is easy to verify that $\phi(\theta, z)$ has a $L_\phi$-Lipschitz continuous gradient [GRW20, Lemma 3] where

$$L_\phi = 2\sqrt{(1 + \beta)^2 + (1 + 1/(2\beta))^2}.$$

Using the definition of $\phi(\theta, z)$ in (51), and Lipschitz continuity of its gradient, we have

$$\phi(\tilde{\theta}_k, \tilde{z}_k) - \phi(\tilde{\theta}_{k+1}, \tilde{z}_{k+1})$$

$$\leq \left\langle \tilde{z}_k + \beta(\tilde{y}_k - \tilde{\theta}_k), \tilde{\theta}_{k+1} - \tilde{\theta}_k\right\rangle - \left\langle \tilde{y}_k - \tilde{\theta}_k, \tilde{z}_{k+1} - \tilde{z}_k\right\rangle + \frac{L_\phi}{2}\left[\left\|\tilde{\theta}_{k+1} - \tilde{\theta}_k\right\|_2^2 + \|\tilde{z}_{k+1} - \tilde{z}_k\|_2^2\right]. \tag{52}$$

By the optimality condition of the subproblem (31) we have,

$$\left\langle \tilde{z}_k + \beta(\tilde{y}_k - \tilde{\theta}_k), y - \tilde{y}_k\right\rangle \geq 0 \quad \forall y \in \Theta. \tag{53}$$

For $y = \tilde{\theta}_k$ we have,

$$\left\langle \tilde{z}_k + \beta(\tilde{y}_k - \tilde{\theta}_k), \tilde{y}_k - \tilde{\theta}_k\right\rangle \leq 0. \tag{54}$$

Note that this also implies

$$\phi(\tilde{\theta}_k, \tilde{z}_k) \leq 0. \tag{55}$$

We also have,

$$
\begin{aligned}
&\tilde{z}_{k+1} - \tilde{z}_k \\
=&\tilde{z}_{k+1} - (1 - \eta_{k+1})\tilde{z}_k - \eta_{k+1}\tilde{z}_k \\
=&z_{k+1} - (1 - \eta_{k+1})z_k - \eta_{k+1}\tilde{z}_k + \tilde{\zeta}_{k+2} - (1 - \eta_{k+1})\tilde{\zeta}_{k+1} \\
=&\eta_{k+1}\left(\nabla f(\theta_k) + e_{k+1} + \nu_{k+1} + \zeta_{k+1}\right) - \eta_{k+1}\tilde{z}_k + \tilde{\zeta}_{k+2} - (1 - \eta_{k+1})\tilde{\zeta}_{k+1} \\
=&\eta_{k+1}\left(\nabla f(\theta_k) + e_{k+1} + \nu_{k+1}\right) + (\tilde{\zeta}_{k+1} - \tilde{\zeta}_{k+2}) - \eta_{k+1}\tilde{z}_k + \tilde{\zeta}_{k+2} - (1 - \eta_{k+1})\tilde{\zeta}_{k+1} \\
=&\eta_{k+1}\left(\nabla f(\theta_k) + \tilde{\epsilon}_{k+1}\right) - \eta_{k+1}\tilde{z}_k,
\end{aligned}
$$

where, $\tilde{\epsilon}_k = e_k + \nu_k + \tilde{\zeta}_k$. Then, using (16) we have,

$$
\begin{aligned}
&\left\langle \tilde{y}_k - \tilde{\theta}_k, \tilde{z}_{k+1} - \tilde{z}_k \right\rangle \\
=&\left\langle \tilde{y}_k - \tilde{\theta}_k, \eta_{k+1}\left(\nabla f(\theta_k) + \tilde{\epsilon}_{k+1}\right) - \eta_{k+1}\tilde{z}_k \right\rangle \\
=&\left\langle \tilde{\theta}_{k+1} - \tilde{\theta}_k, \nabla f(\tilde{\theta}_k) \right\rangle + \left\langle \tilde{\theta}_{k+1} - \tilde{\theta}_k, \nabla f(\theta_k) - \nabla f(\tilde{\theta}_k) \right\rangle + \left\langle \tilde{y}_k - \tilde{\theta}_k, \eta_{k+1}\tilde{\epsilon}_{k+1} \right\rangle - \left\langle \tilde{\theta}_{k+1} - \tilde{\theta}_k, \tilde{z}_k \right\rangle \\
\geq& f(\tilde{\theta}_{k+1}) - f(\tilde{\theta}_k) - \frac{L_G}{2}\|\tilde{\theta}_{k+1} - \tilde{\theta}_k\|_2^2 - \frac{\beta}{2\eta_{k+1}}\left\|\tilde{\theta}_{k+1} - \tilde{\theta}_k\right\|_2^2 - \frac{\eta_{k+1}}{\beta}\left\|\nabla f(\theta_k) - \nabla f(\tilde{\theta}_k)\right\|_2^2 \\
&+ \left\langle \tilde{y}_k - \tilde{\theta}_k, \eta_{k+1}\tilde{\epsilon}_{k+1} \right\rangle - \left\langle \tilde{\theta}_{k+1} - \tilde{\theta}_k, \tilde{z}_k \right\rangle. \tag{56}
\end{aligned}
$$

Combining (52), (53), (54), and (56), using (16), and rearranging, we get,

$$
\begin{aligned}
&\phi(\tilde{\theta}_k, \tilde{z}_k) - \phi(\tilde{\theta}_{k+1}, \tilde{z}_{k+1}) \\
\leq& f(\tilde{\theta}_k) - f(\tilde{\theta}_{k+1}) + \frac{L_G}{2}\|\tilde{\theta}_{k+1} - \tilde{\theta}_k\|_2^2 + \frac{\beta}{2\eta_{k+1}}\left\|\tilde{\theta}_{k+1} - \tilde{\theta}_k\right\|_2^2 + \frac{\eta_{k+1}}{\beta}\left\|\nabla f(\theta_k) - \nabla f(\tilde{\theta}_k)\right\|_2^2 \\
&- \left\langle \tilde{y}_k - \tilde{\theta}_k, \eta_{k+1}\tilde{\epsilon}_{k+1} \right\rangle - \eta_{k+1}\beta\left\|\tilde{y}_k - \tilde{\theta}_k\right\|_2^2 + \frac{L_\phi}{2}\left[\left\|\tilde{\theta}_{k+1} - \tilde{\theta}_k\right\|_2^2 + \|\tilde{z}_{k+1} - \tilde{z}_k\|_2^2\right] \\
&W(\tilde{\theta}_{k+1}, \tilde{z}_{k+1}) - W(\tilde{\theta}_k, \tilde{z}_k) \\
\leq& -\frac{\eta_{k+1}\beta}{2}\left\|\tilde{y}_k - \tilde{\theta}_k\right\|_2^2 + \frac{(L_G + L_\phi)\eta_{k+1}^2}{2}\left\|\tilde{y}_k - \tilde{\theta}_k\right\|_2^2 + \frac{L_\phi}{2}\|\tilde{z}_{k+1} - \tilde{z}_k\|_2^2 + \frac{\eta_{k+1}L_G^2}{\beta}\left\|\theta_k - \tilde{\theta}_k\right\|_2^2 \\
&- \eta_{k+1}\left\langle \tilde{y}_k - \tilde{\theta}_k, \tilde{\epsilon}_{k+1} \right\rangle
\end{aligned}
$$

Summing both sides from $k = 0$ to $N - 1$, and using (55), we get,

$$\sum_{k=0}^{i} \frac{\eta_{k+1}\beta}{2}\left\|\tilde{y}_k - \tilde{\theta}_k\right\|_2^2 \leq W(\tilde{\theta}_0, \tilde{z}_0) + \sum_{k=0}^{N-1} r_{k+1},$$

where

$$r_{k+1} = \frac{(L_G + L_\phi)\eta_{k+1}^2}{2}\left\|\tilde{y}_k - \tilde{\theta}_k\right\|_2^2 + \frac{L_\phi}{2}\|\tilde{z}_{k+1} - \tilde{z}_k\|_2^2 + \eta_{k+1}\left\langle \tilde{\theta}_k - \tilde{y}_k, \tilde{\epsilon}_{k+1} \right\rangle + \frac{\eta_{k+1}L_G^2}{\beta}\left\|\theta_k - \tilde{\theta}_k\right\|_2^2.$$

■

*Proof.* [Proof of Lemma A.6] We omit the details of the proof of Lemma A.6 since the proof is similar to Proposition 1 of [GRW20] except that we no longer have $\mathbb{E}\left[(\tilde{\theta}_k - \tilde{y}_k)^\top \tilde{\epsilon}_k | \mathcal{F}_{k-1}\right] = 0$ since $\{\tilde{\epsilon}_k\}_k$ is no longer a martingale difference sequence. But we can show that the term is small enough, i.e., of the order of the stepsize. Note that, using (41), we have

$$
\begin{aligned}
\mathbb{E}\left[(\tilde{\theta}_k - \tilde{y}_k)^\top \tilde{\epsilon}_k | \mathcal{F}_{k-1}\right] =& \mathbb{E}\left[(\tilde{\theta}_k - \tilde{y}_k)^\top (\nu_k + \tilde{\zeta}_k) | \mathcal{F}_{k-1}\right] \\
\leq& \sqrt{\mathbb{E}\left[\|\tilde{\theta}_k - \tilde{y}_k\|_2^2 | \mathcal{F}_{k-1}\right]} \sqrt{\mathbb{E}\left[\|\nu_k + \tilde{\zeta}_k\|_2^2 | \mathcal{F}_{k-1}\right]}
\end{aligned}
$$

$$\leq \sqrt{\mathbb{E}\left[\frac{2\|\tilde{z}_k\|_2^2}{\beta^2}|\mathcal{F}_{k-1}\right]}\eta_k$$

$$\leq \frac{2\sigma\eta_k}{\beta}. \tag{57}$$

Combining (57) with Proposition 1 of [GRW20] we get Lemma A.6. ∎

## A.3 Auxilliary Results

Let

$$Y_i^i = \mathbf{I} \qquad Y_i^k = \prod_{j=i+1}^{k}\left(1 - \frac{\eta_j}{4}\right).$$

Then, we have the following results:

**Lemma A.7** *For* $k \leq i \leq N$,

$$\left\|Y_{k-1}^i\right\|_2 \leq \exp\left(-\frac{1}{8}\left((N+k)^{1-a} - (N+i+1)^{1-a}\right)\right)$$

$$\sum_{i=k}^{N}Y_{k-1}^i\eta_i = \mathcal{O}(1). \tag{58}$$

*Proof.* Using the fact that $1 - x \leq \exp(-x)$, we get

$$Y_{k-1}^i = \prod_{j=k}^{i}\left(1 - \frac{\eta_j}{4}\right) \leq \prod_{j=k}^{i}\exp\left(-\frac{\eta_j}{4}\right) = \exp\left(-\sum_{j=k}^{i}\frac{(N+j)^{-a}}{4}\right)$$

$$\leq \exp\left(-\int_{k}^{i}\frac{(N+j)^{-a}}{8}\right)dj$$

$$\leq \exp\left(-\frac{1}{8}\left((N+i)^{1-a} - (N+k)^{1-a}\right)\right)$$

Now,

$$\sum_{i=k}^{N}Y_{k-1}^i\eta_i$$

$$\leq \sum_{i=k}^{N}\exp\left(-\frac{1}{8}\left((N+i)^{1-a} - (N+k)^{1-a}\right)\right)(N+i)^{-a}$$

$$= \exp\left(\frac{(N+k)^{1-a}}{8}\right)\sum_{i=k+N}^{2N}\exp\left(-\frac{i^{1-a}}{8}\right)i^{-a}$$

$$\leq \exp\left(\frac{(N+k)^{1-a}}{8}\right)\int_{k+N}^{2N}\exp\left(-\frac{(i-1)^{1-a}}{8}\right)(i-1)^{-a}di$$

$$= \frac{\exp\left(\frac{(N+k)^{1-a}}{8}\right)}{1-a}\int_{(k+N-1)^{1-a}}^{(2N-1)^{1-a}}\exp\left(-u\right)du$$

$$\leq \frac{\exp\left(\frac{(N+k)^{1-a}}{8} - \frac{(N+k-1)^{1-a}}{8}\right)}{1-a}$$

$$\leq \frac{e}{1-a}$$

∎

# B  Proof of Theorem 3.2

Before introducing the main proof we present some notations. Recall that,

$$\phi(\theta, z) = \min_{y \in \Theta} \left( \langle z, y - \theta \rangle + \frac{\beta}{2} \|y - \theta\|_2^2 \right),$$

, and,

$$y_k' = \operatorname*{argmin}_{y \in \Theta} \left\{ \langle z_k, y - \theta_k \rangle + \frac{\beta}{2} \|y - \theta_k\|_2^2 \right\}.$$

For a given $\theta$, and $z$, we introduce the following notation for convenience.

$$H(y) = \langle z, y - \theta \rangle + \frac{\beta}{2} \|y - \theta\|_2^2$$

Then we have [Jag13],

$$\frac{\beta}{2} \|y_k - y_k'\|_2^2 \le H(y_k) - H(y_k').$$

We choose the parameters of Algorithm 2 such that

$$H(y_k) - H(y_k') \le \delta_k^2.$$

We will choose $\delta_k$ later. Let us define the following merit function.

$$W(\theta, z) = (f(\theta) - f^*) - \phi(\theta, z) + \alpha \|\nabla f(\theta) - z\|_2^2. \quad \alpha > 0 \tag{59}$$

We need the following result from [AMP05] on mixing properties of the data under Assumption 2.4 (a).

**Lemma B.1** *[AMP05] Let Assumption 2.4 (a) be true. Then, for any $\theta \in \Theta$, the chain $\{x_k\}_k$ is exponentially mixing in the sense of Definition 25.*

*Proof.* First we establish recursion relations on the three components of $W(\theta, z)$: $(f(\theta) - f^*)$, $\phi(\theta, z)$, and $\alpha \|\nabla f(\theta) - z\|_2^2$.

Using (16), Assumption 2.1, Young's inequality,

$$
\begin{aligned}
&f(\theta_{k+1}) - f(\theta_k) \\
&\le \nabla f(\theta_k)^\top (\theta_{k+1} - \theta_k) + \frac{L_G}{2} \|\theta_{k+1} - \theta_k\|_2^2 \\
&= \eta_{k+1} \nabla f(\theta_k)^\top (y_k' - \theta_k) + \eta_{k+1} (\nabla f(\theta_k) - z_k)^\top (y_k - y_k') + \eta_{k+1} (z_k + \beta(y_k' - \theta))^\top (y_k - y_k') \\
&\quad - \eta_{k+1} \beta \langle y_k' - \theta_k, y_k - y_k' \rangle + \frac{L_G \mathcal{D}_\Theta^2 \eta_{k+1}^2}{2} \\
&\le \eta_{k+1} \left( H(y_k) - H(y_k') - \frac{\beta}{2} \|y_k - y_k'\|_2^2 \right) + \frac{\eta_{k+1} \beta}{16} \|y_k' - \theta_k\|_2^2 + 4 \eta_{k+1} \beta \|y_k - y_k'\|_2^2 + \frac{L_G \mathcal{D}_\Theta^2 \eta_{k+1}^2}{2} \\
&\quad + \frac{\eta_{k+1} \beta}{16} \|\nabla f(\theta_k) - z_k)\|_2^2 + \frac{4 \eta_{k+1}}{\beta} \|y_k - y_k'\|_2^2 + \eta_{k+1} \nabla f(\theta_k)^\top (y_k' - \theta_k) \\
&\le \eta_{k+1} (H(y_k) - H(y_k')) + \frac{\eta_{k+1} \beta}{16} \|y_k' - \theta_k\|_2^2 + 4 \eta_{k+1} \beta \|y_k - y_k'\|_2^2 + \frac{L_G \mathcal{D}_\Theta^2 \eta_{k+1}^2}{2} \\
&\quad + \frac{\eta_{k+1} \beta}{16} \|\nabla f(\theta_k) - z_k)\|_2^2 + \frac{4 \eta_{k+1}}{\beta} \|y_k - y_k'\|_2^2 + \eta_{k+1} \nabla f(\theta_k)^\top (y_k' - \theta_k). \tag{60}
\end{aligned}
$$

Using (52),

$$
\begin{aligned}
&\phi(\theta_k, z_k) - \phi(\theta_{k+1}, z_{k+1}) \\
&\le \langle z_k + \beta(y_k' - \theta_k), \theta_{k+1} - \theta_k \rangle - \langle y_k' - \theta_k, z_{k+1} - z_k \rangle + \frac{L_\phi}{2} \left[ \|\theta_{k+1} - \theta_k\|_2^2 + \|z_{k+1} - z_k\|_2^2 \right] \\
&\le \eta_{k+1} \langle z_k + \beta(y_k' - \theta_k), y_k' - \theta_k \rangle + \eta_{k+1} \langle z_k + \beta(y_k' - \theta_k), y_k - y_k' \rangle - \langle y_k' - \theta_k, z_{k+1} - z_k \rangle
\end{aligned}
$$

$$+ \frac{L_\phi}{2} \left[ \|\theta_{k+1} - \theta_k\|_2^2 + \|z_{k+1} - z_k\|_2^2 \right]$$

$$\leq \eta_{k+1} \left( H(y_k) - H(y_k') - \frac{\beta}{2} \|y_k - y_k'\|_2^2 \right) - \eta_{k+1} \langle y_k' - \theta_k, \nabla F(\theta_k, x_{k+1}) \rangle$$

$$+ \eta_{k+1} \langle y_k' - \theta_k, z_k \rangle + \frac{L_\phi}{2} \left[ \|\theta_{k+1} - \theta_k\|_2^2 + \|z_{k+1} - z_k\|_2^2 \right]$$

$$\leq - \eta_{k+1}\beta \|y_k' - \theta_k\|_2^2 + \eta_{k+1} \left( H(y_k) - H(y_k') \right) - \eta_{k+1} \langle y_k' - \theta_k, \nabla f(\theta_k) \rangle$$

$$- \eta_{k+1} \langle y_k' - \theta_k, \xi_{k+1}(\theta_k, x_{k+1}) \rangle + \frac{L_\phi}{2} \left[ \|\theta_{k+1} - \theta_k\|_2^2 + \|z_{k+1} - z_k\|_2^2 \right]. \tag{61}$$

Recall $\Gamma_i$ defined in (44). Then

$$\nabla f(\theta_i) - z_i = \frac{\Gamma_i}{\Gamma_{i-1}} \left( \nabla f(\theta_{i-1}) - z_{i-1} \right) + \left( \nabla f(\theta_i) - \nabla f(\theta_{i-1}) \right) - \eta_i \left( \tilde{\epsilon}_i + \nabla f(\theta_{i-1}) - \nabla f(\theta_{i-1}) \right)$$

$$= (1 - \eta_i) \left( \nabla f(\theta_{i-1}) - z_{i-1} \right) + \frac{\eta_i}{\eta_i} \left( \nabla f(\theta_i) - \nabla f(\theta_{i-1}) \right) - \eta_i \xi_i.$$

Using Jensen's inequality,

$$\|\nabla f(\theta_i) - z_i\|_2^2 \leq (1 - \eta_i) \|\nabla f(\theta_{i-1}) - z_{i-1}\|_2^2 + \frac{1}{\eta_i} \|\nabla f(\theta_i) - \nabla f(\theta_{i-1})\|_2^2 + \eta_i^2 \|\xi_i\|_2^2$$

$$- 2\eta_i \langle \xi_i, (1 - \eta_i) \left( \nabla f(\theta_{i-1}) - z_{i-1} \right) + \left( \nabla f(\theta_i) - \nabla f(\theta_{i-1}) \right) \rangle$$

$$\leq (1 - \eta_i) \|\nabla f(\theta_{i-1}) - z_{i-1}\|_2^2 + 2L_G^2 \eta_i \|y_{i-1}' - \theta_{i-1}\|_2^2 + 2L_G^2 \eta_i \|y_{i-1} - y_{i-1}'\|_2^2 + \eta_i^2 \|\xi_i\|_2^2$$

$$- 2\eta_i \langle \xi_i, (1 - \eta_i) \left( \nabla f(\theta_{i-1}) - z_{i-1} \right) + \left( \nabla f(\theta_i) - \nabla f(\theta_{i-1}) \right) \rangle. \tag{62}$$

Now combining (60), (61), and (62) we have,

$$W(\theta_{k+1}, z_{k+1}) - W(\theta_k, z_k)$$

$$= f(\theta_{k+1}) - f(\theta_k) - \phi(\theta_{k+1}, z_{k+1}) + \phi(\theta_k, z_k) + \alpha \|\nabla f(\theta_{k+1}) - z_{k+1}\|_2^2 - \alpha \|\nabla f(\theta_k) - z_k\|_2^2$$

$$\leq 2\eta_{k+1} \left( H(y_k) - H(y_k') \right) - \frac{15\alpha\eta_{k+1}}{16} \|\nabla f(\theta_k) - z_k\|_2^2 - \left( \frac{15\beta\eta_{k+1}}{16} - 2\alpha L_G^2 \eta_{k+1} \right) \|y_k' - \theta_k\|_2^2$$

$$+ \eta_{k+1} \left( 4\beta + 4/\alpha + 2L_G^2 \alpha \right) \|y_k - y_k'\|_2^2$$

$$+ \eta_{k+1}^2 \left( \frac{L_G D_\Theta^2}{2} + \frac{L_\phi D_\Theta^2}{2} + \|z_k - \nabla F(\theta_k, x_{k+1})\|_2^2 + \|\xi_{k+1}(\theta_k, x_{k+1})\|_2^2 + 2 \|\xi_{k+1}(\theta_k, x_{k+1})\|_2 \|\nabla f(\theta_k) - z_k\|_2 \right)$$

$$- \eta_{k+1} \langle y_k' - \theta_k, \xi_{k+1}(\theta_k, x_{k+1}) \rangle - 2\eta_{k+1} \langle \xi_{k+1}(\theta_k, x_{k+1}), \nabla f(\theta_{k+1}) - z_k \rangle$$

Rearranging, and choosing $\alpha = \beta/(32L_G^2)$ we get,

$$\frac{14\beta\eta_{k+1}}{16} \|y_k' - \theta_k\|_2^2 + \frac{15\beta\eta_{k+1}}{512L_G^2} \|\nabla f(\theta_k) - z_k\|_2^2$$

$$\leq W(\theta_k, z_k) - W(\theta_{k+1}, z_{k+1}) + \left( 4/\beta + 4\beta + 4/\alpha + 2L_G^2 \alpha \right) \eta_{k+1} \delta_k^2 + \eta_{k+1}^2 U_k - \eta_{k+1} S_k - \eta_{k+1} Q_k, \tag{63}$$

where

$$U_k = \frac{L_G D_\Theta^2}{2} + \frac{L_\phi D_\Theta^2}{2} + \|z_k - \nabla F(\theta_k, x_{k+1})\|_2^2 + \|\xi_{k+1}(\theta_k, x_{k+1})\|_2^2 + 2 \|\xi_{k+1}(\theta_k, x_{k+1})\|_2 \|\nabla f(\theta_k) - z_k\|_2,$$

$S_k = \langle y_k' - \theta_k, \xi_{k+1}(\theta_k, x_{k+1}) \rangle$, and $Q_k = 2 \langle \xi_{k+1}(\theta_k, x_{k+1}), \nabla f(\theta_{k+1}) - z_k \rangle$. Taking expectation on both sides and summing from $k = 0$ to $k = N$, we get,

$$\sum_{k=0}^{N} \mathbb{E} \left[ \frac{14\beta\eta_{k+1}}{16} \|y_k' - \theta_k\|_2^2 + \frac{15\beta\eta_{k+1}}{512L_G^2} \|\nabla f(\theta_k) - z_k\|_2^2 \right]$$

$$\leq W(\theta_0, z_0) + \sum_{k=0}^{N} \left( 4/\beta + 4\beta + 4/\alpha + 2L_G^2 \alpha \right) \eta_{k+1} \delta_k^2 + \sum_{k=0}^{N} \eta_{k+1}^2 \mathbb{E}[U_k] - \sum_{k=0}^{N} \eta_{k+1} (\mathbb{E}[S_k] + \mathbb{E}[Q_k]), \tag{64}$$

**Bound on** $\mathbb{E}[U_k]$ : Similar to (41), we have $\mathbb{E}[\|z_k\|_2] \le \sigma$. Using Lipschitz continuity of $f(\cdot)$, as explained in Section 2, we have $\nabla f(\theta_k) \le L$. Combining these with Assumption 2.3, we have,

$$\mathbb{E}[U_k] = \mathcal{O}(1) \tag{65}$$

**Bound on** $\mathbb{E}[S_k]$ : Using $\mathbb{E}_{x \sim \pi}[\langle y'_{k-l} - \theta_{k-l}, \xi_{k+1}(\theta_{k-l}, x)\rangle] = 0$, for $l \in \{1, \cdots, k-1\}$, we have

$$
\begin{aligned}
&\mathbb{E}[S_k | \mathcal{F}_{k-l}] \\
=&\mathbb{E}[\langle y'_k - \theta_k, \xi_{k+1}(\theta_k, x_{k+1})\rangle | \mathcal{F}_{k-l}] - \mathbb{E}[\langle y'_k - \theta_k, \xi_{k+1}(\theta_{k-l}, x_{k+1})\rangle | \mathcal{F}_{k-l}] \\
&+ \mathbb{E}[\langle y'_k - \theta_k, \xi_{k+1}(\theta_{k-l}, x_{k+1})\rangle | \mathcal{F}_{k-l}] - \mathbb{E}[\langle y'_{k-l} - \theta_{k-l}, \xi_{k+1}(\theta_{k-l}, x_{k+1})\rangle | \mathcal{F}_{k-l}] \\
&+ \mathbb{E}[\langle y'_{k-l} - \theta_{k-l}, \xi_{k+1}(\theta_{k-l}, x_{k+1})\rangle | \mathcal{F}_{k-l}] - \mathbb{E}_{x \sim \pi}[\langle y'_{k-l} - \theta_{k-l}, \xi_{k+1}(\theta_{k-l}, x)\rangle] \\
=&\mathbb{E}\left[\left\langle y'_k - \theta_k, \sum_{i=k-l+1}^{k} (\xi_{k+1}(\theta_i, x_{k+1}) - \xi_{k+1}(\theta_{i-1}, x_{k+1}))\right\rangle | \mathcal{F}_{k-l}\right] \\
&+ \mathbb{E}\left[\left\langle \sum_{i=k-l+1}^{k} (y'_i - \theta_i - y'_{i-1} + \theta_{i-1}), \xi_{k+1}(\theta_{k-l}, x_{k+1})\right\rangle | \mathcal{F}_{k-l}\right] \\
&+ \langle y'_{k-l} - \theta_{k-l}, \mathbb{E}[\xi_{k+1}(\theta_{k-l}, x_{k+1}) | \mathcal{F}_{k-l}]\rangle - \langle y'_{k-l} - \theta_{k-l}, \mathbb{E}_{x \sim \pi}[\xi_{k+1}(\theta_{k-l}, x)]\rangle \\
=&\mathbb{E}\left[\|y'_k - \theta_k\|_2 \sum_{i=k-l+1}^{k} \eta_i \|y_{i-1} - \theta_{i-1}\|_2 | \mathcal{F}_{k-l}\right] \\
&+ \mathbb{E}\left[\sum_{i=k-l+1}^{k} (\|z_i - z_{i-1}\|_2 / \beta + 2\|\theta_i - \theta_{i-1}\|_2) \|\xi_{k+1}(\theta_{k-l}, x_{k+1})\|_2 | \mathcal{F}_{k-l}\right] \\
&+ \|y'_{k-l} - \theta_{k-l}\|_2 \|\mathbb{E}[\xi_{k+1}(\theta_{k-l}, x_{k+1}) | \mathcal{F}_{k-l}] - \mathbb{E}_{x \sim \pi}[\xi_{k+1}(\theta_{k-l}, x)]\|_2.
\end{aligned}
\tag{66}
$$

Using Assumption 2.1 one has,

$$\mathbb{E}\left[\|y'_k - \theta_k\|_2 \sum_{i=k-l+1}^{k} \eta_i \|y_{i-1} - \theta_{i-1}\|_2\right] = \mathcal{O}(l\eta_{k-l+1}). \tag{67}$$

Using Assumption 2.1, Assumption 2.3, $z_{k+1} - z_k = \eta_{k+1}(\nabla F(\theta_k, x_{k+1}) - z_k)$, $\mathbb{E}[\|z_k\|_2] \le \sigma$ one has,

$$\mathbb{E}\left[\sum_{i=k-l+1}^{k} (\|z_i - z_{i-1}\|_2 / \beta + 2\|\theta_i - \theta_{i-1}\|_2) \|\xi_{k+1}(\theta_{k-l}, x_{k+1})\|_2\right] = \mathcal{O}(l\eta_{k-l+1}). \tag{68}$$

Using Assumption 2.1, Assumption 2.4, Lemma B.1, (25), and Lipschitz continuity of $f(\cdot)$, we have,

$$\|y'_{k-l} - \theta_{k-l}\|_2 \|\mathbb{E}[\xi_{k+1}(\theta_{k-l}, x_{k+1}) | \mathcal{F}_{k-l}] - \mathbb{E}_{x \sim \pi}[\xi_{k+1}(\theta_{k-l}, x)]\|_2 \le \mathcal{O}(exp(-rl)), \tag{69}$$

where $r$ is as in (25). Combining (67), (68), and (69) with (66) we get,

$$\mathbb{E}[S_k] = \mathcal{O}(l\eta_{k-l+1} + exp(-rl)) \tag{70}$$

**Bound on** $\mathbb{E}[Q_k]$**:** Following similar techniques used to establish bound on $\mathbb{E}[S_k]$, we have,

$$\mathbb{E}[Q_k] = \mathcal{O}(l\eta_{k-l+1} + exp(-rl)) \tag{71}$$

Combining (65), (70), and (71) with (63), choosing $t_k = \lceil \sqrt{k}\rceil$ to ensure $\delta_k^2 = \eta_{\|}$, setting $l = \lceil \frac{\log(1/\eta_{k-l+1})}{r}\rceil$, and choosing $\eta_k = (N+k)^{-a}$, $1/2 < a < 1$, we get,

$$\sum_{k=0}^{N} \mathbb{E}\left[\frac{14\beta\eta_{k+1}}{16}\|y'_k - \theta_k\|_2^2 + \frac{15\beta\eta_{k+1}}{512L_G^2}\|\nabla f(\theta_k) - z_k\|_2^2\right] \le W(\theta_0, z_0) + \mathcal{O}\left(N^{1-2a}\log N\right),$$
$$\tag{72}$$

Dividing both sides by $\sum_{k=0}^{N} \eta_k$, and choosing $a = 1/2$ we get,

$$\mathbb{E}[V(\theta_R, z_R)] = \mathcal{O}\left(\frac{\log N}{\sqrt{N}}\right).$$

∎