# OpenReview forum: "Constrained Stochastic Nonconvex Optimization with State-dependent Markov Data"
_NeurIPS.cc/2022/Conference — NeurIPS 2022 Accept_

### Official Review · Reviewer_cX4H · 2022-07-10

**Rating:** 5
**Confidence:** 3
**Soundness:** 3 good
**Presentation:** 3 good
**Contribution:** 2 fair

**Summary:**

In this work, the authors consider a class of constrained stochastic optimization problems where the data is drawn from a Markov chain. They propose a conditional gradient-type algorithm for solving this class of problems with bounds on the convergence rate. Some numerical experiments are conducted to verify the effectiveness of the algorithm.

**Questions:**

1. See weakness points.
2. Compared with the literature, the main difficulty seems to be the Markovian data. However, by assumption 2.4 and Proposition 2.1, it seems that the problem can be reduced to classical constrained smooth stochastic optimization. Does it mean that assumption 2.4 is too strong? Or is there any other technical difficulty?

**Strengths And Weaknesses:**

Strength:
1. Constrained stochastic optimization problems with Markovian data are important in the ML community.
2. The theoretical results are complete.
3. The paper is well-written and easy to follow.

Weakness:
1. The word "projection-free" is a bit misleading, as the proposed algorithm still contains a projection step which is solved by the conditional gradient method. And using the conditional gradient method to compute an approximate projection point is well-known in the literature.
2. The experiment is not meaningful enough. LMO has complexity O(n) on the one-norm ball, and projection has complexity O(nlogn). So there is not much computational gain. From the plot, it seems that projection can actually give more stable results. I think it makes more sense to consider, for example, nuclear norm ball, where the complexity for LMO and projection are very different.

---

> ### Author Response · Authors · 2022-08-02
> **Thank you for your valuable comments.**
>
> 1. Please note that the the sub-problem to obtain $y_k$ in Algorithm 1, could be either solved using projection-based solvers (in which case it will be a projection-based algorithm) or assuming access to Linear Minimization Oracles (in which case it will be a projection-free algorithm). For the later case, we quantify the number of calls to the LMO and we never use any projections. Hence, if we assuming and using access to the LMO, the overall algorithm is projection-free.
>
> In fact, we do not claim that projection-free aspect to be the main contribution of this paper (Line 110 - 118). We want to **highlight** here that **analysis of constrained nonconvex optimization for state-dependent Markovian data is the main contribution**. Our analysis is applicable in a unified manner to both the projection-based and projection-free settings.
>
> 2. **Nuclear-norm ball constraint experiment:** Taking the reviewer's comment into account, we have **added  a new experiment in the Appendix C on single-index model regression with a nuclear-norm constraint on the model parameter** (highlighted in blue).
>
> 3. **Theoretical challenge** The main theoretical difficulty is indeed Markovian data although it cannot be reduced to stochastic smooth optimization analysis by Proposition 2.1 alone.
>
> Assumption 2.4 is important although for a different reason. Assumption 2.4 is needed to prove Lemma 3.1. This noise decomposition is crucial to the analysis. The decomposition combined with the auxiliary iterate technique (Line 230 - 231) is the crux of the theory used in this paper.
>
> Please reach out to us in the discussion phase if there are additional questions and we will be happy to provide the required clarification.
>
> [1] Tesi Xiao, Krishnakumar Balasubramanian, and Saeed Ghadimi. A projection-free algorithm for constrained stochastic multi-level composition optimization. arXiv preprint arXiv:2202.04296, 2022.

---

> > ### Author Response · Authors · 2022-08-07
> > **Looking forward to the feedback**
> >
> > Dear Reviewer,\
> > We have answered the questions you raised and added additional experiments taking your comment into consideration. Given that we are nearing the end of the discussion period, we would greatly appreciate if you get back to us and let us know if the answers are satisfactory. If you have any additional questions, we would be happy to answer them.
> >
> > We look forward to hearing from you.  Thank you.
> >
> > Sincerely\
> > Authors.

---

> > > ### Comment · Reviewer_cX4H · 2022-08-08
> > > **Thank you for the reply.**
> > >
> > > Thank you for the reply. I have changed my score.

---

> > > > ### Author Response · Authors · 2022-08-08
> > > > **Thank you for the update**
> > > >
> > > > Thank you for the update.  Please let us know if you have any other concerns and we will be happy to clarify.
> > > >
> > > > Sincerely \
> > > > Authors

---

### Official Review · Reviewer_iPV7 · 2022-07-10

**Rating:** 6
**Confidence:** 3
**Soundness:** 3 good
**Presentation:** 2 fair
**Contribution:** 2 fair

**Summary:**

The paper considers constrained nonconvex optimization under a state-dependent Markov data setting where at each iteration, data samples are generated to a Markov chain that depends on the current parameter. Two algorithms have been analyzed and convergence rates are provided.

**Questions:**

1. The presentation is not so good. The author prefers to cite without explanation. For example, the author didn’t explain what is $L$ in (7). Before presenting Assumption 2.1 and 2.2, they first cite them in Proposition 1.1. The same issue happens with the introduction of Algorithm 1. In Definition 1, the definition of ϵ-stationary solution replies on another unexplained $z$. All of these make Section 1.2 hard to digest.

2. The condition $\mathbb{E}\left[\left\|\nabla F\left(\theta_{k}, x_{k+1}\right)\right\|_{2}^{2} \mid \mathcal{F}_{k}\right] \leq \sigma_{2}^{2}$ in Assumption 2.3 is a kind of uniformly bounded gradient assumption. Could it be relaxed to a bounded gradient assumption at the optimal point? After all, you have assume (13).

3. Could the author give more explanation on the choice of step sizes in (18)? Typical choice of step sizes should be proportion to $t^{-1/2}$ or $t^{-1}$. The used exponent $-3/5$ is quite strange and no explanation is provided, while in Theorem 3.2, the used exponent follows my knowledge. The step size $t^{-a} (0.5<a<1)$ is often used to derive statistical properties and often has inferior last-iterate convergence (which can be fixed by a simple average). Does it imply the inferior oracle complexity under state-dependent MCs is due to the used step sizes?

4. The paper says the proposed setting includes reinforcement learning (RL) as a special case. So could we use the mentioned methods to find the optimal policy in RL? Notice that the value function of a policy is also a non-convex but smooth function of the policy. What’s more, when we use the greedy policy with respect to the current estimator of value function to do exploration, we also meet a stochastic approximation problem with state-dependent Markov data.

5. Is there any lower bound analysis on the number of LMO oracles under state-dependent Markov data? I am not sure whether the proposed is optimal, though the author says the results match previous results in the case of state-independent Markovian data.

**Limitations:**

See the above sections.

**Strengths And Weaknesses:**

Originality: The key technique is to decompose the stochastic error by using the Poisson equation associated with the underlying Markov chain. The technique is not new and has been used previously in many optimization or statistics papers. The main novelty from my view is to apply the “old” technique in the “new” optimization setting and to fill the blank of theories.

Clarity: The paper is overall well-written.  However, some parts are somewhat hard to follow.

Quality: The paper seems to be sound with all theoretical claims supported, though I didn’t check very carefully. But I feel they are correct. The experiment setting is clearly described.

Significance: The paper fills the blank of theories in optimization under Markovian data. It is related to the ML community and might inspire future work.

---

> ### Author Response · Authors · 2022-08-02
> **Thank you for your valuable comments.**
>
> 1. Thank you for pointing this out. We have added some clarification and updated the draft (highlighted in blue). In the camera-ready version, we will rearrange this section adding since we will have one more page.
>
> 2. We respectfully disagree. The condition $\mathbb{E}\left[\left||\nabla F\left(\theta_{k}, x_{k+1}\right)\right||~2^{2} \mid \mathcal{F}{k}\right] \leq \sigma_{2}^{2}$ means that the **conditional variance** of the random variable $\nabla F\left(\theta_{k}, x_{k+1}\right)$ is bounded but **does not imply an uniform bound** on $||\nabla F\left(\theta_{k}, x_{k+1}\right)||$. For example, if $\nabla F\left(\theta_{k}, x_{k+1}\right)$ is multivariate Gaussian random variable then the variance is bounded but not the random variable itself. If we set $\theta'=\theta^*$ in (13) then we get, $||\nabla F(\theta,x)||_2\leq ||\nabla F(\theta^*,x)||_2+c||\theta-\theta^*||_2\mathcal{V}(x)$. Note that even if $||\nabla F(\theta^*,x)||_2$ is bounded, the right hand side of the above equation is not uniformly bounded due to the presence of $\mathcal{V}(x)$.
>
> 3. **What happens for $1/2<a<1$** For $1/2<a<1$, the rate of convergence is given by $\max(N^{a-1},N^{2-4a})$ as in (48). In Theorem 3.1, we chose the $a$ which provides the fastest rate of convergence for clarity. We have updated our draft and stated it as Remark 2 below Theorem 3.1 (highlighted in blue).
>
> 4. Yes. The RL example is indeed an example that would fit in our setup. However, in line with general results in non-convex optimization, our results work for any objective function that satisfies all the mentioned assumptions. Outside of RL, there are also a lot of other problems that would potentially fit in out setup.
>
> 5. We are not aware of any lower bound on the LMO complexity for state-dependent Markovian data. In the state-independent case, we *show* that the rate matches the previous results for *iid* data in [1] except for logarithmic terms. These logarithmic terms are unavoidable since optimization with Markovian data is slower than  *iid* data even in the unconstrained strongly-convex setting [2].  The authors of [3] obtains the same rate in the state-independent Markov setting for constrained nonconvex optimization.
>
> [1] Xiao, Tesi, Krishnakumar Balasubramanian, and Saeed Ghadimi. "A Projection-free Algorithm for Constrained Stochastic Multi-level Composition Optimization." arXiv preprint arXiv:2202.04296 (2022).
>
>
> [2] Nagaraj, Dheeraj, Xian Wu, Guy Bresler, Prateek Jain, and Praneeth Netrapalli. "Least squares regression with markovian data: Fundamental limits and algorithms." Advances in neural information processing systems 33 (2020): 16666-16676.
>
>
> [3] Alacaoglu, Ahmet, and Hanbaek Lyu. "Convergence and Complexity of Stochastic Subgradient Methods with Dependent Data for Nonconvex Optimization." arXiv preprint arXiv:2203.15797 (2022).

---

> > ### Comment · Reviewer_iPV7 · 2022-08-05
> > **Thanks for your resposes.**
> >
> > I have read the authors' responses. Most of my concerns have been addressed. I want to further ask some questions.
> >
> > 1. What if we additionally have $\sup_{x} \mathcal{V} (x) < \infty$? Would the additional assumption help relax Assumption 2.3? I just feel the uniformly bounded $\sigma_2$ in Assumption 2.3 is a little bit strong and we can make use of (13).
> >
> > 2. In your mind, what is the main difficulty in applying the technique from [AMP05] to analyze the mentioned algorithms? The way to handle constrained set?
> >
> > I currently want to maintain my score and wait to see other reviewers' responses.

---

> > > ### Author Response · Authors · 2022-08-05
> > > **Thank you for the feedback.**
> > >
> > > 1. Assuming $\sup_x\mathcal{V}(x)<\infty$ would imply that the gradient is uniformly bounded and one no longer needs Assumption 2.3. Indeed $\sup_x\mathcal{V}(x)<\infty$ is a stronger assumption than assumption 2.3 since $\sup_x\mathcal{V}(x)<\infty$ implies Assumption 2.3 but the converse is not true.
> > >
> > > 2. The main concept that we leverage from the work of [AMP05], is obtaining the decomposition of the noise in Lemma 3.1 for Algorithm 1. We emphasize that we use the word concept here because the error in our Algorithm 1 is different from that of the SGD algorithm considered in [AMP05] because of the moving-average. Apart from this, in our analysis the main challenge is to develop the auxiliary Markov chain (20)-(24) based on this decomposition and show that these iterates are close to the original one. Our overall approach provides an unified framework to handle both the constrained and unconstrained settings.
> > >
> > > Please let us know if you have additional questions and we will be happy to answer them.

---

> > > > ### Author Response · Authors · 2022-08-08
> > > > **Thank you for the feedback.**
> > > >
> > > > Dear Reviewer, Given that the discussion period ends tomorrow and there were several technical interactions with the other reviewers, we were wondering if you have any additional questions or changes in your score (we recall that you had mentioned **I currently want to maintain my score and wait to see other reviewers' responses**).  Thank you!
> > > >
> > > > Sincerely\
> > > > Authors

---

> > > > > ### Comment · Reviewer_iPV7 · 2022-08-09
> > > > > **Keep my score**
> > > > >
> > > > > I am satisfied with the authors' responses and I want to keep my score.

---

### Official Review · Reviewer_6vHk · 2022-07-11

**Rating:** 6
**Confidence:** 2
**Soundness:** 3 good
**Presentation:** 3 good
**Contribution:** 3 good

**Summary:**

The paper studies conditional gradient methods for stochastic optimization with state-dependent markovian data, where the expectation is taken over the stationary distribution of the random vector, utilizing a moving average gradient estimator. The paper established non-asymptotic gradient complexity and linear minimization oracle complexity.

**Questions:**

## Questions:
1. What is the available data? Is it a sequence of state-dependent Markovian data given at the beginning, or is each sample in the sequence given online? I assume the latter.
2. What is the relationship between the convergence criteria of gradient mapping, the Frank- Wolfe gap, and the performative prediction criteria used in [DX20]

## Minor Comments:
1. Lack of space in equation (11)


**Ethics Review Area:**

["I don’t know"]

**Limitations:**

N.A.

**Strengths And Weaknesses:**

## Strengths
1. It is the first non-asymptotic result of its kind of conditional gradient method for stochastic optimization with state-dependent markovian data.
2. The use of a moving average estimator well suits the Markovian data situation and could shed light on other gradient-based methods for state-dependent markovian data.


## Weaknesses
1. Assumption 2.4 is relatively strong as there exists a stationary distribution for any $\theta$.
2. The motivation of Algorithm 1 is less explained. Making it hard to extend the algorithmic design idea to other settings.

---

> ### Author Response · Authors · 2022-08-02
> **Thank you for your valuable comments.**
>
> **On Assumption 2.4** Please note that Assumption 2.4 is satisfied in the performative prediction (strategic classification problem that we use as our motivating example [7].
>
> Furthermore, Assumption 2.4 is fairly common in controlled Markov chain literature [2 - 4]. There are several settings where such an  assumption holds, for example stochastic approximation implementation of Expectation Maximization algorithm [5] and stochastic approximation algorithms to estimate signals received over a noisy channel by minimizing squared loss  in digital communication (See Section 1.2.1.1 of Part I, Section 2.2, and 2.4 of Part II of [6]).
>
> **Motivation of the Algorithm design** The main idea of the algorithm is similar to stochastic projected gradient descent except that we do not use a mini-batch based gradient estimator. In contrast we generate a moving average based estimator $z_k$ of the gradient. This is essential for the analysis in the Markov setting.
>
> To relate to a familiar algorithm like SGD, please note that in the unconstrained setting, one has $y_k=\theta_k-z_k$ which gives $\theta_{k+1}=\theta_k-\eta_{k+1}z_k$.
>
> Questions:
>
> 1. Yes. It is indeed a sequence of state-dependent Markov chain data received in an online manner.
>
> **Relation between criteria in [DX20] and gradient-mapping and Frank-Wolfe** [DX20] considers performative prediction in a strongly-convex unconstrained setting. For unconstrained setting, gradient mapping $||\mathcal{G}_{\Theta}(\theta, \nabla f(\theta), \beta)||^2$ translates to $||\nabla f(\theta)||_2^2$ (line 76). Furthermore, $\mu$-strong convexity implies
> $$||\nabla f(\theta)||_2^2\geq 2\mu(f(\theta)-f(\theta^*))\geq \mu^2||\theta-\theta^*||_2^2,$$
>  where $\theta^*$ is the unique minimizer of $f$. So the performance measure used in [DX20] is upper-bounded by $||\nabla f(\theta)||_2^2$. Similarly, in strongly-convex unconstrained setting, Frank-Wolfe Gap upper bounds $f(\theta)-f(\theta^*)$ [1].
>
> [1] Jaggi, Martin. "Revisiting Frank-Wolfe: Projection-free sparse convex optimization." In International Conference on Machine Learning, pp. 427-435. PMLR, 2013.
>
> [2] Yuksel, Serdar, and Sean P. Meyn. "Random-time, state-dependent stochastic drift for Markov chains and application to stochastic stabilization over erasure channels." IEEE Transactions on Automatic Control 58, no. 1 (2012): 47-59.
>
> [3] Yüksel, Serdar, and Tamer Başar. "Control over noisy forward and reverse channels." IEEE Transactions on Automatic Control 56, no. 5 (2010): 1014-1029.
>
> [4] Andrieu, Christophe, Vladislav B. Tadić, and Matti Vihola. "On the stability of some controlled Markov chains and its applications to stochastic approximation with Markovian dynamic." The Annals of Applied Probability 25, no. 1 (2015): 1-45.
>
> [5] Andrieu, Christophe, and Matti Vihola. "Markovian stochastic approximation with expanding projections." Bernoulli 20, no. 2 (2014): 545-585.
>
> [6] Benveniste, Albert, Michel Métivier, and Pierre Priouret. Adaptive algorithms and stochastic approximations. Vol. 22. Springer Science \& Business Media, 2012.
>
> [7] Li, Qiang, and Hoi-To Wai. "State dependent performative prediction with stochastic approximation." In International Conference on Artificial Intelligence and Statistics, pp. 3164-3186. PMLR, 2022.

---

> > ### Author Response · Authors · 2022-08-08
> > **Looking forward to your feedback**
> >
> > Dear Reviewer, as we are nearing the end of the discussion period, we were wondering if you got a chance to look at our response. We would greatly appreciate if you could let us know if you have any additional questions. Thank you.
> >
> > Sincerely \
> > Authors

---

> > > ### Comment · Reviewer_6vHk · 2022-08-09
> > > **Thanks for the detailed explanation**
> > >
> > > Thanks for the detailed explanation and pointing out those references. I have no further questions.

---

### Official Review · Reviewer_VfkJ · 2022-07-11

**Rating:** 4
**Confidence:** 3
**Soundness:** 2 fair
**Presentation:** 3 good
**Contribution:** 2 fair

**Summary:**

The paper studied a stochastic conditional gradient method for nonconvex constrained optimization under the Markovian sampling setting, where the data comes from a Markov chain. They provide the complexity result of stochastic first-order oracle (SFO) and linear minimization oracle (LMO) for such an algorithm for achieving epsilon-stationary point.

**Questions:**

1. The idea of constructing an auxiliary Markov chain for bounding Markovian sampling error has already been well studied in various unconstrained problems [1] [2] [3]. Hence, it seems to me that the analysis of the proposed algorithm is a (somehow direct) combination of the theory of the conditional gradient method and the technique for dealing with Markovian sampling.

2. The problem setting, stochastic constrained problems with state-dependent Markov chain dynamics in the training data, is not well motivated by the example. [LW22] considers an unconstrained problem, why is it necessary to add a constant like $\|\theta\|_1 \leq R$?

3. The experiments are tiny. In addition, the author claim that the projection-free algorithm is computationally cheaper than the projection-based algorithms. However, this is not presented in the experiment. The algorithm based on projection indeed enjoys better empirical iteration complexity compared with ICG. I would expect the author to provide a time comparison.


[1]  Doan, T. T., Nguyen, L. M., Pham, N. H., & Romberg, J. (2020). Finite-time analysis of stochastic gradient descent under markov randomness. arXiv preprint arXiv:2003.10973.

[2]  Xu, T., Zou, S., & Liang, Y. (2019). Two time-scale off-policy TD learning: Non-asymptotic analysis over Markovian samples. Advances in Neural Information Processing Systems, 32.

[3]  Wu, Y. F., Zhang, W., Xu, P., & Gu, Q. (2020). A finite-time analysis of two time-scale actor-critic methods. Advances in Neural Information Processing Systems, 33, 17617-17628.


**Ethics Review Area:**

["I don’t know"]

**Limitations:**

Yes

**Strengths And Weaknesses:**

-Strengths

The work considers the nonconvex constrained optimization problem under Markovian sampling, and they provide the corresponding complexity results.

-Weakness

See comments below.

---

> ### Author Response · Authors · 2022-08-02
> **Thank you for your valuable comments.**
>
> 1. **Constructing an auxiliary Markov chain** We respectfully point out to the reviewer that among the 3 papers provided as a reference, only [3] uses an auxiliary sequence in the analysis. Furthermore, [2] and [3] develop and analyze two-time scale algorithms whereas our algorithm is a single-time scale algorithm. Hence, we point out that out algorithm and analysis are entirely different from the works provided in the review. Below, we make specific comparisons to the individual papers.
>
>
>  Paper [1] makes that restrictive assumption that the state-space is finite and the transition kernel of the Markov chain is state-independent Markov chain, and it is geometric mixing. We do not make such restrictive assumption.
>
> Paper [2] assumes geometric ergodicity of the chain as well.
>
> Paper [3] indeed deals with state-dependent Markov chain and constructs  an auxiliary Markov chain as a part of their analysis. But the auxiliary chain used in [3] is quite different from ours. Specifically, in [3] the auxiliary chain is generated by repeatedly applying policy $\pi_{\theta_{t-\tau}}$ from state $s_{t-\tau}$ whereas we generate the auxiliary updates as in (20)-(23). More specifically, we generate the chain by perturbing the original $\{z_k\}$  sequence by  $\tilde{\zeta}_{k+1}$ component of the noise. Moreover, the results obtained in [3] is applicable to reinforcement learning problem whereas we treat a general class of nonconvex optimization problem. Furthermore, [3] considers unconstrained problems, whereas our analysis is applicable to unconstrained problems and constrained problems (solved using projection-based or projection-free methods) in an unified manner.
>
>
> 2. **Requirement of $\|\theta_1\|_1\leq R$ constraint**  $\|\theta_1\|_1\leq R$ constraint has been widely used to achieve sparsity in classical statistics literature (see Chapter 3 in [5] for example) as well as neural network classifier literature [1-4] for interpretability. Specifically, in our example, through sparse weight learning, one would like to see which features have more influence on the output after propagating through the network.
>
> We have stated that $\|\theta_1\|_1\leq R$ has been used in this work as an sparsity inducing classifier in Line 64, and Line 275. We will try to make it clearer.
>
> 3. **Experiment on Nuclear-norm ball constraint** We have **included an experiment in Appendix C on matrix-data based single-index model regression with a nuclear-norm constraint on the model parameter** (highlighted in blue) since the projection on nuclear norm-ball is much more costly than the conditional gradient step [6-8]. Nevertheless, we want to emphasize that the main contribution of this paper is theoretical analysis of Inexact Averaged Stochastic Approximation for nonconvex constrained optimization with state-dependent Markov chain data.
>
> [1] Li, Gen, Yuantao Gu, and Jie Ding. "The Efficacy of $ L_1$ Regularization in Two-Layer Neural Networks." arXiv preprint arXiv:2010.01048 (2020).
>
> [2] Y. Cheng, D. Wang, P. Zhou, and T. Zhang. A survey of model compression and acceleration for deep neural networks. arXiv preprint arXiv:1710.09282, 2017.
>
> [3] L. Zhao, Q. Hu, and W. Wang. Heterogeneous feature selection with multi-modal deep neural networks and sparse group LASSO. IEEE Trans. Multimed., 17(11):1936–1948, 2015.
>
> [4] W. Wen, C. Wu, Y. Wang, Y. Chen, and H. Li. Learning structured sparsity in deep neural networks. Advance. Neural Inf. Process. Sys., pages 2074–2082, 2016.
>
> [5] T. Hastie, R. Tibshirani, and J. Friedman. The elements of statistical learning: data mining, inference, and prediction. Springer Science and Business Media, 2009.
>
> [6] Jaggi, Martin. "Revisiting Frank-Wolfe: Projection-free sparse convex optimization." In International Conference on Machine Learning, pp. 427-435. PMLR, 2013.
>
> [7] Martin Jaggi and Marek Sulovsk`y. A simple algorithm for nuclear norm regularized problems. In ICML, 2010.
>
> [8] Harchaoui, Zaid, Anatoli Juditsky, and Arkadi Nemirovski. "Conditional gradient algorithms for norm-regularized smooth convex optimization." Mathematical Programming 152, no. 1 (2015): 75-112.

---

> > ### Comment · Reviewer_VfkJ · 2022-08-06
> > **Thanks for the response**
> >
> > I would like to thank the authors for their response. My concerns remain.
> >
> > - I read the discussions between the authors and the reviewer iPV7. I cannot appreciate the technical novelty of this manuscript at the current stage (the authors claimed that this is a theoretical paper, I evaluate it through its technical novelty). The main difficulty is to invoke Markovian sampling rather than i.i.d. in the algorithm. I think the proof strategy is similar to what is presented in the literature (through reading their proof sketch below Theorem 3.1). To summarize, it is to evaluate the "distance" of the distribution at the current iterate to the stationary distribution. This can be seen from the construction of auxiliary sequence $(\widetilde \theta_k, \widetilde y_k, \widetilde z_k)$ and the construction (24) where they finally regard the stochastic gradient sampled in the Markovian way as that from the stationary distribution with error. This shares nearly the same idea as the analysis of [1]. However, I have to acknowledge that I cannot really understand the ICG method presented in this paper. I know a little about frank-wolfe, but I found it so hard to understand Algorithm 2  (perhaps it is due to my limited understanding of the conditional gradient method). There is no explanation of the algorithm design idea. Thus, it is possible that the main difficulty is invoking the standard Markovian analysis in ICG, even though this is not clearly stated in the proof sketch.
> >
> > - Consider SGD for smooth nonconvex stochastic problems, the complexity for $\mathbb E ||\nabla f(x)||^2$ is $O(\varepsilon^{-2})$. Why the obtained result for the projection method is $O(\varepsilon^{-2.5})$? I tend to think it is due to your techniques for addressing the Markovian sampling since the same method with i.i.d. sampling can achieve the same complexity as SGD (indicated in your Table 1). However, as far as I know, the Markovian sampling often leads to an additional $\log(\varepsilon^{-2})$ rather than an $\varepsilon^{-0.5}$. Is it because you use the techniques from the much older [AMP05] rather than from the more recent [1]?
> >
> > - I did not find your motivating example in [DX20] and [LW22]. Putting the Markovian sampling scheme in an algorithm that is not analyzed in this setting previously always creates "new" results. But I want to know whether it should be appreciated by applications. The results for the unconstrained case are extensively studied. The new part of this manuscript is to consider hard-to-project constraints. However, why do we need to add the $\ell_1$-norm ball constraint? Personally, I think that it cannot be justified by saying $\ell_1$ promotes sparsity, and it should always be a good choice. Even if one really wants to add sparsity, why not add it as a regularizer? It is easy to perform proximal SGD in this case.
> >
> > - The constraint set is assumed to be bounded in this paper. Could you please explain why you need this assumption?
> >
> > - The second assumption in Assumption 2.3 is not good. It is related to certain Lipschitz-type assumption on $F$ itself, which is a hard assumption. It is used in (40) of Lemma A.6. Does the unconstrained work needs this assumption? (SGD does not, as I know).
> >
> > I want to clarify that, by technical novelty, I do not mean that the derivation needs to be tedious. However, if you put your work as some theoretical outcome, it should at least convey some insights of the techniques you used (this is something that will be helpful for others). I cannot appreciate too much if one applies quite standard techniques to a new algorithm to create "new" results, especially when the application of this algorithm is not so clear.
> >
> > I mentioned that I could not understand Algorithm 2 (it is not fully my responsibility as there is no explanation about algorithm design). It is truly possible that I do miss something in the ICG part. I will also inform AC about this factor to let AC consider my evaluation in a careful way.

---

> > > ### Comment · Reviewer_VfkJ · 2022-08-06
> > > **Missed reference**
> > >
> > > Sorry that I missed the reference.
> > >
> > > [1] Wu, Y. F., Zhang, W., Xu, P., & Gu, Q. (2020). A finite-time analysis of two time-scale actor-critic methods. Advances in Neural Information Processing Systems, 33, 17617-17628.

---

> > > ### Author Response · Authors · 2022-08-07
> > > **Thanks for the feedback**
> > >
> > > Thanks for the feedback. Below, we respond to the **factual questions** raised, and clarify several **incorrect** claims made. We also refer the paper [1] (i.e., the paper: A finite-time analysis of two time-scale actor-critic methods) as [D3] for convenience. **Due to character constraint we are breaking up our reply over multiple posts.**
> > >
> > > **"...I cannot appreciate the technical novelty of this manuscript... This shares nearly the same idea as the analysis of [1]:"** Because [D3] has been brought up again, we address the differences between [D3]) and our work in more detail:
> > >
> > > 1. **In [D3], the transition operator $\mathcal{P}$ is state-independent** whereas in **our work $P_\theta$ depends on $\theta$**. It is crucial to allow for this state-dependence for this setup to be applicable to problems like strategic classification with adapted best response. From technical aspect, this state-dependence implies, we no longer have exponential mixing.
> > >
> > >  2. **Regarding using auxiliary Markov chain** We want to emphasize here that except for the fact that [D3] and our work depends on **some** auxiliary Markov chain for analysis, there is no similarity between the two auxiliary chains generated. In [D3], the chain is generated by repeatedly applying policy $\pi_{t-\tau}$ to state $s_{t-\tau}$. If one applies an analogous version of this technique to our setting, that amounts to generating samples by repeatedly applying Markov transition with transition operator $P_{\theta_i}$ to the sample $x_i$ for some $i\geq 0$, and running the algorithm with those hypothetical samples. Specifically, in that case, the hypothetical chain would be, for $k\geq i$,
> > >
> > >
> > >  $y_k'=\min_{y\in\Theta}\left\lbrace \langle z_k',y-\theta_k'\rangle+\frac{\beta}{2}||y-\theta_k'||_2^2\right\rbrace$
> > >
> > >
> > > $\theta_{k+1}' = \theta_{k}'+\eta_{k+1} (y_k'-\theta_k')$
> > >
> > >
> > > $z_{k+1}'=(1-\eta_{k+1})z_{k}'+\eta_{k+1}\nabla F(\theta_k,x_{k+1}'),$
> > >
> > >
> > > where $x_{k+1}'$ is the sample generated from $x_i$ after $k+1-i$ transitions according to transition operator $P_{\theta_i}$. Whereas we generate the chain in (20)-(24) by adding a time varying perturbation to the original set of iterates $z_k$. A comparison with the above iterates with (20)-(24) reveals the difference between our approach and that in [D3]. Moreover [D3] itself relies on the chain generated by a different paper [D13]. **Relying on an auxiliary chain for analysis is common in Markov literature but design of the specific chain varies from problem to problem and the main novelty lies there**.
> > >
> > > **There is no explanation of the algorithm design idea** For algorithm 1, the moving averaging technique helps to avoid increasing batchsize. One important issue in stochastic optimization is to control the variance associated with the gradient estimators. Even in thy iid setting when the problem is convex or unconstrained nonconvex, we can control the summation of the error terms by only assuming the boundedness of the variance/second moment of gradient estimators. However, when the problem in constrained and nonconvex, we have an additional error term when updating the main iterate and thus, we need to reduce the variance of gradient estimators. Taking mini-batch of samples is a variance-reduction technique that has its own limitations such as not being appropriate for online learning. Using the history of generated stochastic gradients and forming a convex combination of them as gradient estimators is another technique to reduce the variance. Our algorithm uses this moving average technique in the Markov setting.
> > >
> > > Regarding the ICG algorithm, it is exactly the Frank-Wolfe algorithm applied to solve the sub-problem required to obtain $y_k$. The reason why the word inexact is used because, we only need an approximate solution after running $t$ steps.
> > >
> > > If you are not aware of the details of Frank-Wolfe algorithm, you could ignore the projection-free part in Algorithm 1 and think of any projection-based (or even proximal version) algorithm to obtain $y_k$. As mentioned in lines 210-211, all we need is an approximate solution to obtain $y_k$ such the error is of order $\mathcal{O}(\eta_k)$.
> > >
> > > **"Markovian setting often leads to an additional $\log(\epsilon^{-2})$ rather than an $\epsilon^{-0.5}$.."** This statement is **not precise**. It is **true under exponential mixing condition**. Note, however, that in our **state-dependent case, exponential-mixing does not hold**.  The $\epsilon^{-2.5}$ rate has also been obtained in reinforcement learning literature in the unconstrained setting with state-dependent policies [D3].For the **state-independent case we indeed obtain  the $\log N/\sqrt{N}$ as shown in Theorem 3.2.** Furthermore, [D1] proves that even with state-independent Markov data, in the unconstrained strongly-convex setting, optimization is slower in terms of convergence than optimization with *iid* data.

---

> > > ### Author Response · Authors · 2022-08-07
> > > **Thanks for the feedback**
> > >
> > > **"I did not find your motivating example in [DX20] and [LW22]"** Please see section 1.2 of [DX20] and Section 2.1, and 5 of [LW22]. In fact our example is the nonconvex classifier version of the exact classification problem described in Section 2.1, and 5 of [LW22].
> > >
> > > **"Even if one really wants to add sparsity ... It is easy to perform proximal SGD in this case."** We highlight that using regularized version of the problem and using proximal algorithms to solve them need not necessarily be computationally easier in general, e.g., in the nuclear norm example we have added in the revision, computing proximal operator, as suggested by you, with the nuclear norm still requires computation of full singular value decomposition (see section 6.7.3 of [D9] for example). However, using LMO only requires to compute the leading left and right singular vectors which is much faster to compute.
> > >
> > > **"The constraint set is assumed to be bounded in this paper."** This is needed to establish stability of the iterates. This is a fairly common assumption for constrained optimization [D6 - D8]. In the unconstrained optimization with Markov chain data, to achieve stability, one resorts to either a bounded function assumption ([D3], [D8]), or modifying the canonical stochastic optimization algorithm with a varying truncation step [D4]. It is possible to append the varying truncation scheme to Algorithm 1, and use recent modifications of conditional gradient methods [D5] to relax the boundedness assumption.
> > >
> > > **"The second assumption in Assumption 2.3 is not good. It is related to certain Lipschitz-type assumption on $F$ itself, ..Does the unconstrained work needs this assumption? (SGD does not, as I know)."** This claim that ``It is related to certain Lipschitz-type assumption on $F$" is incorrect.  This condition does not imply Lipschitz property for $F$ as this requires only the conditional second moment of the random variable $||\nabla F(\theta_k,x_{k+1})||$ to be bounded (as we have explained in our reply to reviewer iPV7). In fact, in this setting, this condition is implied by the bounded noise variance assumption (which is a common assumption across SGD literature; see [D10]-[D12]):
> > >
> > > $ E[||\nabla F(\theta_k,x_{k+1})||_2^2 | \mathcal{F}_k] \leq 2 || \nabla f ( \theta_k) ||_2^2 + 2 E [ || \xi _{k+1} ( \theta _k, x _{k+1} ) ||_2^2 | \mathcal{F} _k ] \leq 2L^2 + 2 \sigma _1^2.$
> > >
> > > The last inequality follows from the Lipschitz continuity property of $f$ ($L$ is mentioned in line 178-179). Lipschitz continuity property of $f$ follows from the fact that any continuously differentiable function (Assumption 2.2) on a compact set (Assumption 2.1) is Lipschitz continuous.
> > >
> > > We believe the confusion might have arisen because perhaps your intuitions are based on unconstrained optimization, but our paper is focused on constrained optimization.

---

> > > > ### Comment · Reviewer_VfkJ · 2022-08-07
> > > > **Thanks for the response**
> > > >
> > > > - Assumption 2.3. I meant certain Lipschitz-type assumption. In your derivation, you use the Lipschitz assumption of $f$. I meant bounded gradient/subgradent is often equivalent to Lipschitz continuity, thus calling the expected version **Lipschitz-type**. Now, line 178-179 reminds me that everything lives in a compact domain. This assumption is justified by the compact domain assumption, which, I would say, is not a light one. The authors also mentioned "only the conditional second moment" and "bounded noise variance assumption (which is a common assumption across SGD literature)". The conditional expectation assumptions hold in an almost sure sense. Standard SGD uses (total) expected bounded variance rather than this conditional version. Or?
> > > >
> > > > - State-dependent. I believe that the work [D3] is **state-dependent**, since the change of policy will definitely influence the $P_{\theta}$. While the exponential mixing does not hold under the state-dependent setting due to the change of $\theta$ at each iteration, the sampling error will only be of order $O(\log(\epsilon^{-1}))$ plus an error **which is proportional to square of $\theta$'s stepsize** (can be shown by the Lipschitz of the stationary distribution w.r.t. $\theta$). Thus, the state-dependent issue will only influence the rate to a logarithmic term. In fact, the $\tilde{O}(\epsilon^{-2.5})$ rate in [D3] is not caused by the state-dependent issue mentioned by authors, but its bi-level problem structure which restricts the stepsize of $\theta$ to be two-timescale.

---

> > > > > ### Author Response · Authors · 2022-08-07
> > > > > **Thanks for the feedback**
> > > > >
> > > > > 1a) The compactness assumption on the constrained set is very common in the analysis of conditional gradient algorithms (see, e.g., [DA1 - DA4]).
> > > > >
> > > > > 1b) We were responding to the point **The second assumption in Assumption 2.3 is not good** that you mentioned and the connection to the unconstrained SGD setting.
> > > > >
> > > > > Specifically, we mentioned that given the constrained setting that we are working  with (including the compactness assumption) the second assumption in Assumption 2.3 is immediately satisfied.  Again, we mentioned this because you were asking for connections to the unconstrained SGD.
> > > > >
> > > > > In the light of the above two points, we would like to re-emphasize that your intuitions seem to be rather based on **unconstrained** stochastic optimization and we kindly request you to reconsider your stance on our work which is on **constrained** stochastic optimization. [D3] is a two-time scale algorithm that is focusing specifically on a particular **unconstrained** RL problem. Our work is a single-timescale algorithm focusing on a general class of **constrained** nonconvex problems with state-dependent Markovian data. Furthermore, as we pointed out previously our auxiliary sequence is entirely different compared to [D3]. Hence, we believe it is extremely unfair to be dismissive of our work based on [D3].
> > > > >
> > > > > **While the exponential mixing ... due to the change of $\theta$ at each iteration, the sampling error will only be of order $O(\log(\epsilon^{-1}))$ plus an error which is proportional to square of $\theta$'s stepsize (can be shown by the Lipschitz of the stationary distribution w.r.t. $\theta$)**
> > > > >
> > > > > We are not sure how you obtain " the sampling error will only be of order $O(\log(\epsilon^{-1}))$ plus an error which is proportional to **square** of $\theta$'s stepsize" based on Lipschitz property of the stationary distribution. If you could kindly provide a formal proof or specific pointer to a theorem from the literature we would be in a better position to check the required assumptions ands answer this concern. Note that our current result is true based on the provided proof.
> > > > >
> > > > > We reemphasize that our focus is (a) based on a **single observed trajectory** of a Markov chain and (b) on **single time-scale algorithms**. To our understanding your statement  "sampling error will only be of order $O(\log(\epsilon^{-1}))$ plus an error which is proportional to **square** of $\theta$'s stepsize" would be true if one could apply the same transition operator $P_{\theta_k}$ keeping $\theta_k$ fixed at least for a few iteration. This leads to a minibatch based approach where you can generate several samples at each $\theta_k$. This violates point (a) above. Morally speaking, in [D3] the sequence $\theta_k$ is sufficiently fixed from $\omega_k$'s perspective because of two-time scale nature of the algorithm - indeed, for the faster iterates,  the slower iterates behave as though they are sufficiently fixed. This is indeed the main motivation of two-time scale algorithms (see the paragraph after eq (1.3) in [DA5] for example). However, as mentioned in point (b), our focus is on single time-scale algorithms.
> > > > >
> > > > > [DA1]  Jaggi, Martin. "Revisiting Frank-Wolfe: Projection-free sparse convex optimization." In International Conference on Machine Learning, pp. 427-435. PMLR, 2013.
> > > > >
> > > > > [DA2] Lan, Guanghui, and Yi Zhou. "Conditional gradient sliding for convex optimization." SIAM Journal on Optimization 26, no. 2 (2016): 1379-1409.
> > > > >
> > > > > [DA3] Qu, Chao, Yan Li, and Huan Xu. "Non-convex conditional gradient sliding." In International Conference on Machine Learning, pp. 4208-4217. PMLR, 2018.
> > > > >
> > > > > [DA4] Yurtsever, Alp, Suvrit Sra, and Volkan Cevher. "Conditional gradient methods via stochastic path-integrated differential estimator." In International Conference on Machine Learning, pp. 7282-7291. PMLR, 2019.
> > > > >
> > > > > [DA5] Konda, Vijay R., and John N. Tsitsiklis. "Convergence rate of linear two-time-scale stochastic approximation." The Annals of Applied Probability 14, no. 2 (2004): 796-819.

---

> > > > > > ### Author Response · Authors · 2022-08-09
> > > > > > **Look forward to your response**
> > > > > >
> > > > > > Dear Reviewer,\
> > > > > > Given that we are hours away from discussion period deadline, we were wondering if you could provide us the requested proof or references from the literature for your claim.
> > > > > >
> > > > > > Please also recall that we have answered all of your other questions, and clarified several incorrect claims made. Unless you have other questions, we would greatly appreciate if you could reconsider your stance on our paper and raise your score to **quantitatively** amd meaningfully reflect the clarifications we have provided to all your questions. Thanks for your consideration.
> > > > > >
> > > > > > Sincerely\
> > > > > > Authors.

---

> > > > > > ### Comment · Reviewer_VfkJ · 2022-08-09
> > > > > > **Thanks for the feedback.**
> > > > > >
> > > > > > I maintain my concerns about the technical novelty. Indeed, several of the authors' replies are not so accurate, e.g., claiming the reference I mentioned is state-independent. Though the authors consider the constrained case, as claimed by the authors, the main difficulty is to bound errors due to state-dependent sampling. Currently, I do not think this is so algorithmic-dependent that invoking this analysis to the conditional gradient is so challenging. I have already recommended [1]. If the authors go through this paper carefully, they should find that (5.5) in [1] bounds the error caused by state-dependent Markovian sampling, which is somehow independent of the algorithms they used. In addition, I do not think the authors addressed the application issue. It is too coarse. Therefore, I maintain my score.
> > > > > >
> > > > > > [1] Wu, Y. F., Zhang, W., Xu, P., & Gu, Q. (2020). A finite-time analysis of two time-scale actor-critic methods. Advances in Neural Information Processing Systems, 33, 17617-17628.

---

> > > > > > > ### Author Response · Authors · 2022-08-09
> > > > > > > **Clarification**
> > > > > > >
> > > > > > > Thank you for the response.
> > > > > > >
> > > > > > > We just wanted to make a quick clarification to regarding another incorrect claim you make now (i.e., "claiming the reference I mentioned is state-independent.").
> > > > > > >
> > > > > > > Please see https://openreview.net/forum?id=xqyDqMojMfC&noteId=Axm_fl1usja we mention "Paper [3] ( changed to [D3] in the later discussions) indeed deals with state-dependent Markov chain"

---

> > > ### Author Response · Authors · 2022-08-07
> > > **References**
> > >
> > > [D1] Nagaraj, Dheeraj, Xian Wu, Guy Bresler, Prateek Jain, and Praneeth Netrapalli. "Least squares regression with markovian data: Fundamental limits and algorithms." Advances in neural information processing systems 33 (2020): 16666-16676.
> > >
> > > [D2] Alacaoglu, Ahmet, and Hanbaek Lyu. "Convergence and Complexity of Stochastic Subgradient Methods with Dependent Data for Nonconvex Optimization." arXiv preprint arXiv:2203.15797 (2022).
> > >
> > > [D3] Wu, Y. F., Zhang, W., Xu, P., and Gu, Q. (2020). A finite-time analysis of two time-scale actor-critic methods. Advances in Neural Information Processing Systems, 33, 17617-17628.
> > >
> > > [D4] Liang, Faming. "Trajectory averaging for stochastic approximation MCMC algorithms." The Annals of Statistics 38, no. 5 (2010): 2823-2856.
> > >
> > > [D5] Wang, Haoyue, Haihao Lu, and Rahul Mazumder. "Frank-Wolfe methods with an unbounded feasible region and applications to structured learning." arXiv preprint arXiv:2012.15361 (2020).
> > >
> > > [D6] Duchi, John C., Alekh Agarwal, Mikael Johansson, and Michael I. Jordan. "Ergodic mirror descent." SIAM Journal on Optimization 22, no. 4 (2012): 1549-1578.
> > >
> > > [D7] Ghadimi, Saeed, Andrzej Ruszczynski, and Mengdi Wang. "A single timescale stochastic approximation method for nested stochastic optimization." SIAM Journal on Optimization 30, no. 1 (2020): 960-979.
> > >
> > > [D8] Zhang, Mingrui, Zebang Shen, Aryan Mokhtari, Hamed Hassani, and Amin Karbasi. "One sample stochastic frank-wolfe." In International Conference on Artificial Intelligence and Statistics, pp. 4012-4023. PMLR, 2020.
> > >
> > > [D9] Parikh, Neal, and Stephen Boyd. "Proximal algorithms." Foundations and trends® in Optimization 1, no. 3 (2014): 127-239.
> > >
> > > [D10] Xu, Yi, Zhuoning Yuan, Sen Yang, Rong Jin, and Tianbao Yang. "On the Convergence of (Stochastic) Gradient Descent with Extrapolation for Non-Convex Minimization." In IJCAI, pp. 4003-4009. 2019.
> > >
> > > [D11] Drori, Yoel, and Ohad Shamir. "The complexity of finding stationary points with stochastic gradient descent." In International Conference on Machine Learning, pp. 2658-2667. PMLR, 2020.
> > >
> > > [D12] Ghadimi, Saeed, and Guanghui Lan. "Stochastic first-and zeroth-order methods for nonconvex stochastic programming." SIAM Journal on Optimization 23, no. 4 (2013): 2341-2368.
> > >
> > > [D13] Zou, S., Xu, T. and Liang, Y. (2019). Finite-sample analysis for sarsa with linear function
> > > approximation. In Advances in Neural Information Processing Systems.

---

### Author Response · Authors · 2022-08-02
**Title modification to reflect the contributions better**

Dear PC, Senior AC, AC and Reviewers,

To better reflect the contributions of the paper, and emphasize that our results are applicable to both projection-based and projection-free algorithms in an unified manner, we have decided to remove the word ``Projection-Free" from the title in our revision. Furthermore, we have also modified the Algorithm 1 and Theorems 3.1 and 3.2 appropriately. Furthermore, we have added a new experiment with trace-norm constraints, as suggested by one of the reviewers in our revision.

Best,

Authors.

---

### Author Response · Authors · 2022-08-06
**Looking forward to the feedback/discussion**

Dear Reviewers VfkJ, 6vHk and cX4H

As we are half way through the discussion period already, we were wondering if you have any further questions based on our initial author response. Please let us know if you have additional questions and we will be happy to answer them.

sincerely,

Authors.

---

### Meta-Review · Area_Chair_4eat · 2022-08-25

**Recommendation:** Accept
**Confidence:** Less certain

**Metareview:**

The paper considers constrained smooth nonconvex problems in a stochastic setting where the data comes from a Markov chain. For this setting, the paper proposes an algorithm that converges to an $\epsilon$-stationary point with stochastic oracle complexity $1/\epsilon^{2.5}$. The paper further shows that in the settings where the projection oracle is expensive to compute (e.g., under nuclear norm constraints), the algorithm can be implemented with $\mathcal{O}(\frac{1}{\epsilon^{5.5}})$ calls to a linear minimization oracle. On the technical side, the principal contributions are in designing a novel moving-average gradient estimator suitable for Markovian data and in designing an auxiliary Markov chain based on a noise decomposition idea (similar to [AMP05]), whose iterates are close to the iterates of the original algorithm. The analysis is sufficiently general to handle both constrained and unconstrained settings.

The motivation for studying the considered problems comes from strategic classification (an example used in the numerical experiments provided in the paper) and reinforcement learning. These problems are of high interest to the ML community and the contributions of the paper seem sufficient. The results will plausibly lead to more developments in this area.

**Award:**

No

---

### Decision · Program_Chairs · 2022-09-14

Accept